# Canopy mortality has doubled in Europe's temperate forests over the last three decades

Cornelius Senf [1,2], Dirk Pflugmacher[1], Yang Zhiqiang[3], Julius Sebald[2], Jan Knorn[1], Mathias Neumann [2], Patrick Hostert [1,4] & Rupert Seidl [2]

Mortality is a key indicator of forest health, and increasing mortality can serve as bellwether for the impacts of global change on forest ecosystems. Here we analyze trends in forest canopy mortality between 1984 and 2016 over more than 30 Mill. ha of temperate forests in Europe, based on a unique dataset of 24,000 visually interpreted spectral trajectories from the Landsat archive. On average, 0.79% of the forest area was affected by natural or human-induced mortality annually. Canopy mortality increased by $+2.40\%$ year$^{-1}$, doubling the forest area affected by mortality since 1984. Areas experiencing low-severity mortality increased more strongly than areas affected by stand-replacing mortality events. Changes in climate and land-use are likely causes of large-scale forest mortality increase. Our findings reveal profound changes in recent forest dynamics with important implications for carbon storage and biodiversity conservation, highlighting the importance of improved monitoring of forest mortality.

[1] Geography Department, Humboldt-Universität zu Berlin, Unter den Linden 6, 10099 Berlin, Germany. [2] Institute for Silviculture, University of Natural Resources and Life Sciences (BOKU) Vienna, Peter-Jordan-Str. 82, 1190 Vienna, Austria. [3] Department of Forest Ecosystems and Society, Oregon State University, Corvallis, OR 97331, USA. [4] Integrated Research Institute on Transformation of Human-Environment Systems (IRI THESys), Humboldt-Universität of Berlin, Unter den Linden 6, 10099 Berlin, Germany. Correspondence and requests for materials should be addressed to C.S. (email: corneliussenf@gmail.com)

Forests ecosystems cover approximately 30% of the Earth's land surface[1]. They provide numerous ecological, economic, and social benefits to humanity, including—but not limited to—supplying timber, purifying water, and serving as places of high recreational and spiritual value[2]. Forests also exert a strong regulating function on global biogeochemical cycles and the climate system[3], sequestering up to 60% of the anthropogenic carbon emissions in recent years[4]. Human well-being thus strongly depends on the state and development of forest ecosystems.

A particularly important ecological process in this context is forest mortality. Tree death is a natural demographic process in forests, and dead and decaying trees are an integral part of healthy forest ecosystems[5]. Standing and downed deadwood, for instance, fosters biodiversity in forests, providing habitat to a variety of species[6]. Also, the early-seral habitats emerging after natural disturbance are diverse and species-rich ecosystems[7,8]. However, elevated levels of tree death can substantially alter ecosystem structure and functioning and impact the manifold services forest ecosystems provide to humanity[9,10]. For example, increased mortality can impact drinking water quality[11] and timber supply[12]. Moreover, elevated mortality decreases the carbon residence time in live biomass and soils[13,14] and could thus substantially reduce the carbon storage potential of forests[15]. Hence, increasing mortality rates are an important indicator of degrading forest health, which in turn could have strong detrimental effects on society[16,17].

Increasing rates of tree death have been reported in recent years for unmanaged[18,19] as well as managed[20] forest ecosystems. These increases have been linked to changes in the climate system in general, and specifically to a higher propensity of extreme climatic events, such as droughts[21,22]. Moreover, growing evidence suggests an amplification of natural disturbances from insects, pathogens, and forest fires under climate change[23,24]. However, the generally increasing levels of atmospheric humidity, $CO_2$ concentration, and the lengthening of the growing seasons with increasing temperatures could also alleviate mortality in many regions[25], as they facilitate primary productivity[26,27]. Furthermore, in many parts of the world tree harvesting by humans and anthropogenic land-use change are the most important causes of forest mortality[28,29]. Anthropogenic factors could either modulate or override climate-induced changes in tree mortality[30,31] or render forests more susceptible to highly climate-sensitive natural disturbances[32,33]. Legacies from past land use thus might significantly contribute to observed changes in forest mortality. Yet, recent developments in forest management toward "close-to-nature" silviculture[34] and large-scale land abandonment[35] might dampen or even reverse impacts from land use on forest mortality. Consequently, there is inconclusiveness about recent trends in forest mortality for many forest ecosystems around the globe.

Large-scale changes in forest mortality remain difficult to detect. Approaches based on dendroecology yield a long-term perspective on forest mortality change[36,37], but inference is highly restricted in space and limited to investigations of remnant old-growth forests. Another important source of information are compilations of gray literature reports on past mortality events[33,38]. These datasets can provide large spatial coverage, but the quality of reporting frequently declines in earlier years, which makes change detection uncertain[33]. A third information source for detecting mortality change is forest inventory data, enabling the analysis of mortality at the level of individual trees[18,19]. However, systematic analyses of mortality change based on inventory data are frequently hampered by widely varying forest inventory systems between countries. This is particularly limiting in areas with highly diverse administrative systems, as in Europe, where consistent information on large-scale changes in forest mortality is largely lacking.

We here present a consistent and comprehensive analysis of forest mortality change across temperate forests in Europe. We visually interpreted 24,000 satellite-derived spectral time series spanning the period from 1984 to 2016 to quantify canopy mortality change across a forest area of approximately 30 Mill. ha, spanning six different countries in Europe (i.e., Austria, Czechia, Germany, Poland, Slovakia, and Switzerland). We defined canopy mortality rate as the percentage of forest area in which the dominant tree layer experienced a discrete mortality event (i.e., a disturbance from natural or anthropogenic causes) in a given year, assessed at a grain of 30 m pixels. Our key objective was to determine whether canopy mortality has increased across Europe's temperate forests in recent years. We tested the null hypothesis of no temporal trend for a 33-year time series of annual canopy mortality rates. We further scrutinized the variation in temporal trends among countries to illuminate potential drivers of mortality change: Based on previous findings of high climate sensitivity of forest mortality[20,22] and the large-scale synchronization of forest development after World War II[39,40], we hypothesized forest mortality to change consistently across countries. Alternatively, if changes in mortality were primarily driven by local variation in forest management, considerably varying trends between countries could be expected. To further elucidate the role of regional-scale drivers for changing mortality, we regressed mortality trends over important variables of climate and forest structure at the country scale.

Subsequently, we investigated canopy mortality trends separately for stand-replacing events (i.e., retaining no live trees at the level of a 30 m pixel) and non-stand-replacing mortality events (i.e., with residual live trees present after the event). If mortality change is primarily caused by high severity natural disturbances (i.e., cyclonal storm events) and clear-cut harvesting, we would expect to see a stronger response of stand-replacing mortality in our data. Alternatively, if mortality changes are primarily the result of selective natural disturbances (e.g., host tree-specific insect activity) and "close-to-nature" silviculture (i.e., thinning, single tree and/or group selection management, see Brang et al.[34]), the signal of non-stand-replacing mortality would dominate the observed mortality trend.

Finally, we contextualized our results in a multi-proxy analysis of mortality indicators. Theory on change detection in ecosystems suggests that the inferential potential can be substantially increased by jointly studying multiple indicators of change[41]. Hence, in addition to remotely sensed canopy mortality changes, we also analyzed trends in wood removal from official harvesting reports, individual tree mortality estimates from the ICP forest network[22,42], and gray literature estimates of wind and bark beetle disturbances[33]. As these additional datasets report on specific aspects of forest mortality (e.g., wood removals, disturbance) rather than providing a comprehensive account of all mortality causes (as does our satellite-based estimate), a comparison across datasets is not suitable to validate our methodology. It rather provides a multifaceted view of the changes in a crucial ecosystem process and amends the remotely sensed information presented here with regard to its causal interpretation. Based on theoretical considerations on the relationship of canopy, volume, and individual tree mortality (see Supplementary Note 1), in conjunction with increasing growing stock[39] and harvesting intensity[43] across Europe, we expected a high correlation between canopy mortality and wood extraction but a negative trend in individual tree mortality.

## Results

**Canopy mortality rates and trends.** Average canopy mortality rates between 1984 and 2016 were 0.79 (0.75–0.83)% of the total

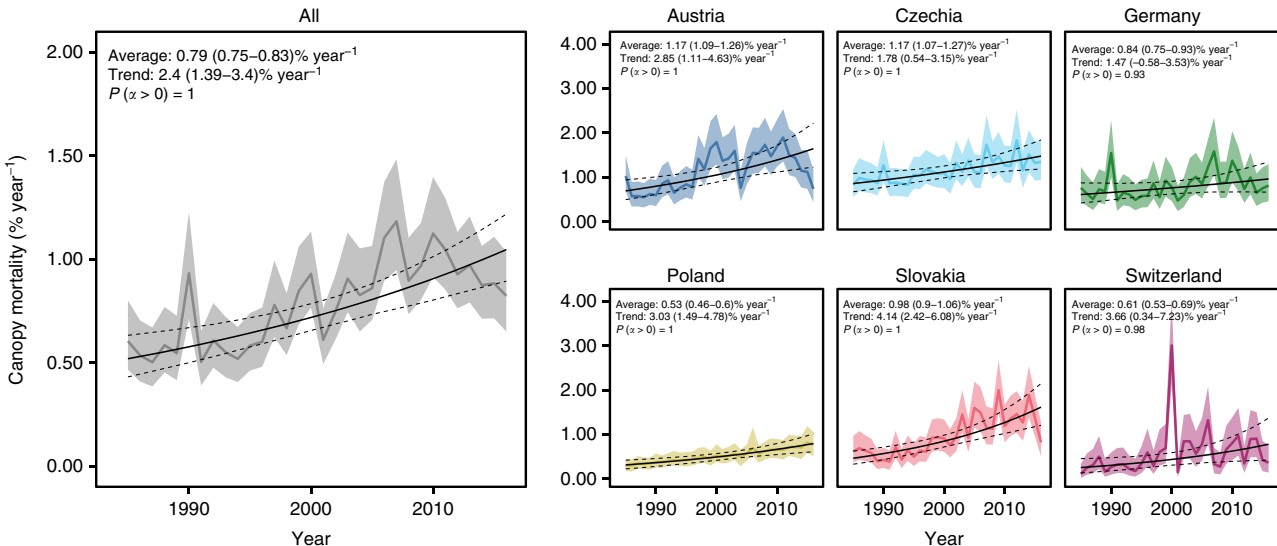

**Fig. 1** Annual rates of canopy mortality in temperate forests of Europe. Estimates were derived from satellite time series interpretation and are reported in percentage of the forest area experiencing canopy mortality. Solid lines indicate the median of the posterior probability distribution. Ribbons and dashed lines indicate the 95% credible interval of the annual estimates and the trend line, respectively

forest area per year (Fig. 1; For all estimates, we report the median of the posterior distribution and the 95% credible interval). Canopy mortality rates varied considerably among countries, ranging from 0.53 (0.46–0.60)% year$^{-1}$ in Poland to 1.17 (1.09–1.26)% year$^{-1}$ in Austria and 1.17 (1.07–1.27)% year$^{-1}$ in Czechia (Supplementary Table 3). There was strong evidence for an increase in canopy mortality rate (indicated as fractional change $\alpha$; see Methods) across temperate forests of Europe over the past 33 years (Fig. 1; $P(\alpha > 0) = 1.00$). Overall, canopy mortality rates increased by 2.40 (1.39–3.40)% year$^{-1}$. Strong evidence for increasing canopy mortality was further found for all individual countries (all $P(\alpha > 0) > 0.98$) with the exception of Germany, where trends were positive but weaker than in other countries ($P(\alpha > 0) = 0.93$). Canopy mortality increase varied from 1.47 (−0.58 to 3.53)% year$^{-1}$ in Germany to 4.14 (2.42–6.08)% year$^{-1}$ in Slovakia (Supplementary Table 3).

**Comparing mortality trends to climate and forest change**. To assess whether observed trends in canopy mortality are related to spatio-temporal variation in forest structure and climate, we compared mortality change in 5-year intervals to two climate variables (mean annual temperature and total annual precipitation) and two attributes of forest structure (growing stock per hectare and median age). We found strong evidence ($P(\beta_1 > 0)$ > 0.99, with $\beta_1$ being the estimated effect) for a positive relationship of mortality with temperature and growing stock (Fig. 2). Canopy mortality increased by 0.41 (0.35–0.47)% with a 1 °C increase in mean annual temperature ($R^2 = 0.38$ [0.15–0.52]), and by 0.66 (0.53–0.80)% with a 100 m$^3$ ha$^{-1}$ increase in growing stock ($R^2 = 0.39$ [0.14–0.49]). A weak positive relationship was also found for total annual precipitation ($R^2 = 0.17$ [0.00–0.39]), while variation in median age was not related to mortality changes ($R^2 = 0.05$ [0.00–0.29]).

**Separating stand-replacing and non-stand-replacing events**. Temporal trends differed significantly between stand-replacing and non-stand-replacing mortality (Fig. 3). Overall, the observed changes in canopy mortality from 1984 to 2016 were strongly driven by increases in non-stand-replacing mortality. Particularly in Czechia and Germany, increases were caused by elevated levels of non-stand-replacing mortality, with a simultaneous tendency of decreasing stand-replacing mortality events. An opposite pattern was observed for Poland, Austria, and Slovakia, which all showed increases in both stand-replacing and non-stand-replacing mortality.

**Multi-proxy analysis of forest mortality**. Comparing satellite-based canopy mortality estimates to other proxies of forest mortality revealed a multifaceted perspective on mortality change in temperate forests of Europe. Remotely sensed canopy mortality rates were strongly correlated with wood removal statistics ($r = 0.75$ [0.65–0.84]; Fig. 4b), with wood removal rates also increasing across temperate forests of Europe (1.4 [0.57–2.29]% year$^{-1}$; Fig. 4a and Supplementary Table 3). In contrast, area-based forest mortality was negatively correlated to estimates of individual tree mortality ($r = -0.33$ [−0.57 to −0.06]; Fig. 4b), with the average rate of individual tree mortality showing a weak and uncertain decrease by −1.45 (−4.31 to 1.36)% year$^{-1}$ over the observation period (Supplementary Table 3). Comparing area-based mortality time series to gray literature data on timber disturbed, we found a moderate positive correlation for bark beetles ($r = 0.27$ [0.12–0.42]; Fig. 4b), whereas the relationship to wind disturbances was weak ($r = 0.16$ [0.01–0.31]; Fig. 4b). Nonetheless, the three large-scale wind storm events in 1990, 1999, and 2007 are also evident in our remotely sensed canopy mortality estimates (Fig. 4a).

**Discussion**

We here present a consistent (across space and time) and comprehensive (i.e., capturing the diverse set of prevailing causes of mortality) assessment of canopy mortality across >30 Mill. ha of temperate forest in Europe. We found that on average 0.79% of the forest area—approximately 240,000 ha—were affected by canopy mortality annually. Furthermore, canopy mortality increased by 2.40% year$^{-1}$, resulting in a doubling of the forest area affected by canopy mortality between 1984 and 2016. Our null hypothesis of no mortality change over time can thus be rejected with a high level of confidence ($P(\alpha > 0) = 1.00$).

The increase in canopy mortality was largely consistent across countries, despite their high variability in forest types and

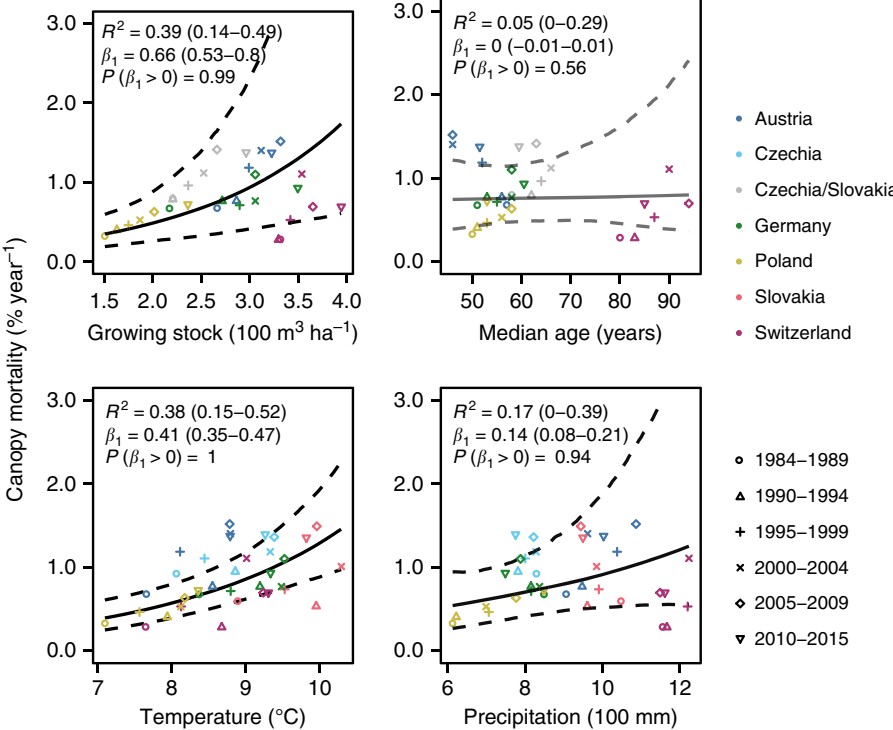

**Fig. 2** Relationship of canopy mortality to spatiotemporal variation in climate and forest structure. Relationships were derived from a linear mixed-effects model with intercept and slope varying among countries. Solid lines indicate the median of the posterior probability distribution. Dashed lines indicate the 95% credible interval. See Supplementary Figure 5 for individual country models

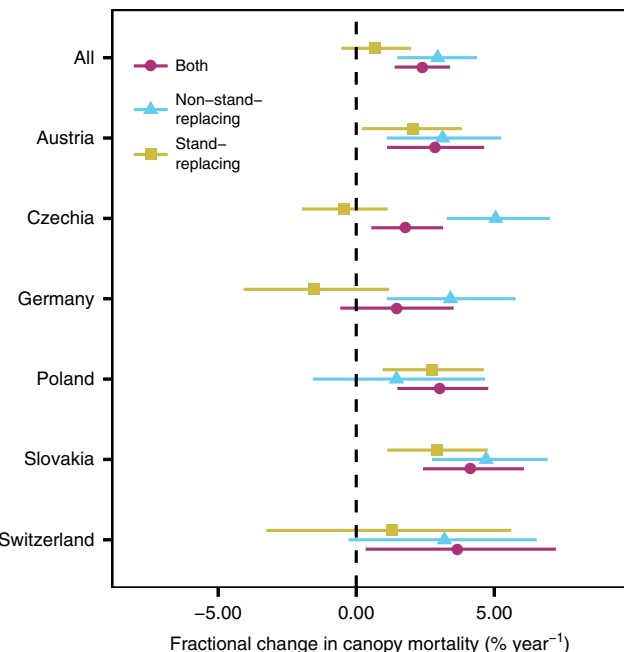

**Fig. 3** Changes in stand-replacing and non-stand-replacing canopy mortality. Mortality is considered stand-replacing if there are no live trees after the mortality event at the level of a 30 m pixel. Points indicate the median of the posterior probability distributions, and bars extent to the 95% credible interval

management systems[44]. This suggests that broader-scale processes such as climate change and forest recovery from past land use—i.e. drivers affecting ecosystems dynamics across national borders and at large spatial scales—are important factors contributing to the observed increases in forest mortality. Our finding of a consistent significant relationship of mortality with increasing temperatures and growing stocks across all countries supports this hypothesis. Both climate change (i.e., increasing mean temperature) and forest recovery from past land use (i.e., increasing growing stocks) are, for instance, important drivers of the prevailing natural disturbance regime in temperate Europe[36,45].

While increasing natural disturbances likely contributed to the observed mortality trend[38], land-use change and specifically intensified tree harvesting—including salvage logging of naturally disturbed areas—is the most important agent of canopy mortality in Europe's temperate forests. This interpretation is supported by a strong correlation of observed canopy mortality trends with reported wood removals (Fig. 4), suggesting that trees are removed for human usage from most of the areas experiencing mortality. Our results indicate that increased extraction is happening primarily in the form of non-stand-replacing mortality (Fig. 3). This suggests increased thinning activity and a transition from past clear-cut systems toward "close-to-nature" silviculture[34] and retention forestry[46] in the temperate forests of Europe. Such an interpretation is also consistent with the weak relationship between forest age and mortality found here (Fig. 2), as these new silvicultural systems move away from simple age-based approaches of stand rotation.

Trends in mortality across our multi-proxy analysis were divergent but generally conformed to theoretical expectations (see Supplementary Note 1), with inventory-based individual tree mortality decreasing over past decades, whereas canopy mortality and official wood removal statistics indicating increasing trends. Our multi-proxy analysis thus suggests that larger forest areas and/or areas of high growing stocks are particularly affected by mortality, with fewer (but bigger) trees dying in these forests. This pattern is consistent with changes in the structure and

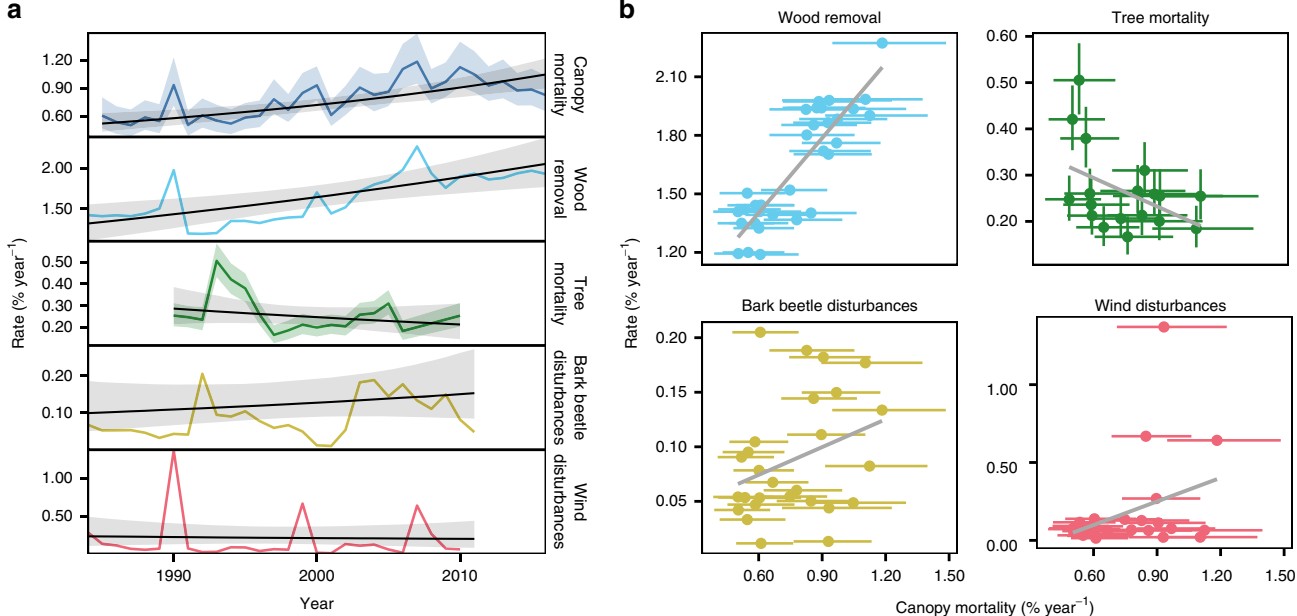

**Fig. 4** A multi-proxy analysis of mortality changes in temperate forests of Europe. Panel **a** shows annual rates (colored lines) and temporal trends (black lines) of canopy mortality (percentage of area affected), wood removal (percentage of growing stock removed), individual tree mortality (percentage of trees dying), bark beetle disturbance (percentage of growing stock disturbed), and wind disturbances (percentage of growing stock disturbed).
Panel **b** shows correlation between canopy mortality (x axis) and the alternate proxies of forest mortality. The gray lines are a linear regression based on the median of the posterior distributions. We note that both wind and bark beetle disturbances are largely included in the total wood removal, due to the common practice of salvage logging in our study area. Solid lines in **a** and dots in **b** are the median of the posterior probability distributions, and ribbons in **a** and error bars in **b** indicate the 95% credible interval. See Supplementary Table 3 for numerical estimates

demography of temperate forests in Europe[39,40] and with empirical relationships between tree size and stem mortality[47]. Moreover, it has important implications for the carbon cycling in Europe's forests: As big trees are crucial for the forest carbon budget and carbon residence times[48], our analysis suggests that the observed reduction in the sink strength of Europe's forests[49] might indeed result from increasing mortality and decreasing carbon residence times[38]. To better quantify the future carbon stocks in Europe's forests, it is thus of paramount importance to improve the representation of mortality processes in simulation models used for policy and decision support[50,51].

In addition to carbon cycle effects, intensifying forest mortality could benefit biodiversity by increasing forest deadwood stocks (which are currently low in the temperate forests of Europe[52]). However, as the increased canopy mortality observed in this study is strongly driven by intensified wood extraction (Fig. 4), deadwood stocks are increasing at a much slower rate than the increase in mortality reported here would suggest[52]. The potential benefits of increasing canopy openness for biodiversity[7,53] might thus be offset by simultaneously increasing wood extraction rates. Beyond the impacts on carbon cycling and biodiversity, the drastic mortality changes reported here impact a wide variety of other ecosystem services[9,10].

Methodologically, our results highlight that a comprehensive analysis of forest mortality change requires a multi-proxy inference across a range of spatial scales[16,54]. The divergent trends in forest mortality between proxies uncovered here (Fig. 4) suggest that the varying responses of forest mortality to climate change reported in the literature might simply be the result of a focus on different response variables. More generally, considering only a single proxy of forest mortality can result in misleading conclusions with respect to underlying drivers of change[55]. We here show that remote sensing holds great potential for analyzing mortality across large areas characterized by heterogeneous

administrative entities. We thus suggest that remote sensing should be at the core of a future forest health monitoring system that captures tree death consistently and comprehensively at the global scale (see also refs [1,54]). Notwithstanding the power of emerging remote-sensing datasets, limitations remain. Our analysis, for instance, only pertains to tree mortality in the canopy layer, underlining the need to complement remote-sensing information with terrestrial data on forest health. Canopy-penetrating, active satellite systems might allow deeper insights into the structural changes associated with canopy mortality[56]. Yet, those systems are currently limited to the analysis of recent mortality events, precluding the quantification of temporal trends.

Europe's forest ecosystems are changing, and mortality is likely to be among the ecosystem processes changing most drastically. A doubling of canopy mortality within 33 years—as observed here—constitutes a substantial change in ecosystem dynamics and has the potential to cancel out simultaneously observed increases in tree growth[26,57]. The drastic changes in forest mortality documented here also highlight the importance of continuously monitoring ecosystem change and utilize this information to foster forest resilience in policy and management decisions[58].

## Methods
**Study area**. Our analysis extends over six countries in Europe (Austria, Czechia, Germany, Poland, Slovakia, and Switzerland), covering a total land area of nearly 93 Mill. ha and a total forest area of approximately 30 Mill. ha. These particular countries were selected because they represent the variability in the temperate forest ecosystems of Europe well (see Supplementary Table 1), cover major bio-physical gradients (lowland to alpine and maritime to continental conditions), and are characterized by a wide range of forest management systems.

**Landsat data**. We downloaded and processed all Landsat data spanning the period between 1984 and 2016 from the United States Geological Survey (USGS) and the

European Space Agency (ESA) archives. Data from the USGS archive were ready-to-use surface reflectance products, while images from the ESA archive needed to be corrected to surface reflectance using LEDAPS[59]. Further, we spatially matched images from the ESA archive to images from the USGS archive using algorithms proposed in Gao et al.[60].

**Sampling design**. We applied a random sampling design to select pixels for visual interpretation of canopy mortality. We decided for a random sampling—i.e., sampling across forest and non-forest areas—as no reliable forest cover map was available for stratification in the beginning of our study period (1984). We subsequently stratified sampling by country (with equal number of samples per country) in order to yield similar precision among countries and to make estimates comparable to other datasets available at the country scale. The number of samples per country was determined to balance data collection effort and precision. After testing varying sample sizes using an existing stand-replacing forest disturbances map for 2000–2012[1], we set the final sample size to 4000 plots per country, resulting in 24,000 samples in total.

**Interpretation and response design**. For interpreting Landsat time series with respect to changes in forest canopy, we followed the TimeSync approach described in Cohen et al.[61]. TimeSync allows a human interpreter to subdivide yearly Landsat time series into linear segments of stable, declining, or increasing spectral trends (Supplementary Figure 1). With the aid of Landsat images and high-resolution imagery available in Google Earth, the interpreter can determine whether spectral changes correspond to forest canopy changes or whether spectral changes were caused by other artifacts, such as phenological variation or cloud cover. Visual Landsat interpretation has been shown to excel at forest change detection relative to automatic algorithms[62]. It furthermore significantly reduces commission error for non-stand-replacing changes and allows the detection of multiple change events within one time series.

To facilitate interpretation, the six spectral bands of Landsat were transformed into Tasseled Cap space[63], which allows for an assessment of vegetation brightness, greenness, and wetness in RBG (red, blue, green) color space. As a first step, the interpreter assessed the land use of the pixel. Definitions of the land uses considered are given in Supplementary Table 2. For all pixels identified as forest, the interpreter subsequently assessed whether and when there has been a change (or multiple changes) in canopy cover over the study period. This was done by inspecting the spectral trajectory for breaks or gradual changes (see Supplementary Figure 1 for an example), while simultaneously inspecting Landsat images and high-resolution imagery (if available). If the spectral change was attributed to a change in canopy cover, the interpreter selected the start and end points of a change period (subsequently referred to as vertices), thus subdividing the time series into linear segments of similar behavior. For each segment, the change process was identified (stable, canopy mortality, regrowth; see Supplementary Table 2). In general, a decrease in Tasseled Cap Wetness is associated with a decrease in canopy cover due to mortality, but the interpreter also was able to assess simultaneous changes in other spectral bands to aid attribution. Finally, land use and land cover was set for the vertices delimiting each segment. We did not separate specific agents of canopy change, as no historic high-resolution imagery was available throughout all countries, potentially introducing a significant attribution bias in earlier years. However, based on the land cover information recorded for each vertex, we distinguished stand-replacing mortality events (i.e., resulting in a non-treed land cover) from non-stand-replacing mortality events (i.e., resulting in a treed land cover). Estimating the percentage of forest cover at the level of a Landsat pixel is inherently difficult. We here used high-resolution imagery to calibrate interpreters in making decisions on whether a mortality event results in a complete loss of forest cover (stand-replacing mortality) or whether residual live trees remained (non-stand-replacing mortality). We subsequently evaluated our classification of stand-replacing and non-stand-replacing mortality events by testing for differences in relative residual forest cover between both classes at 280 plot locations selected from an independent dataset (see Supplementary Methods 1 for details).

**Statistical estimation of mortality rates and trends**. We estimated annual mortality rates using a hierarchical binomial model with logit link function. The full model is described in detail in Supplementary Methods 2. In essence, the model estimates annual mortality rates, with each year's rate being assumed to emerge from the same underlying population. The mean of the binomial distribution is modeled as a linear model with time as predictor, thus estimating the fractional change ($\alpha$) in mortality rate over time. We developed individual models for each country and sampled joint posterior distributions of all parameters using Monte Carlo Markov Chain methods as implemented in Stan (version 2.17.0)[64]. Finally, a mean estimate across all countries was obtained by calculating the weighted mean of the posterior distributions of each country estimate. Weights were calculated proportional to the area of each country, accounting for the stratified sampling design implemented in the TimeSync analysis. We finally summarized all posterior distributions with regard to their median and 95% credible interval (2.5% and 97.5% quantile of the posterior) and calculated the probability of a positive trend ($P(\alpha > 0)$).

**Relating mortality trends to climate and forest change**. To elucidate the relationship of climate and forest structure with changing forest mortality, we regressed forest mortality trends over four covariates, including changes in climate (mean annual temperature and total annual precipitation) as well as forest structure (median age and growing stock density). Variable selection was based on previous analyses on drivers of forest mortality in Europe[45]. Mean annual temperature and total annual precipitation time series were obtained from 2962 weather stations through the European Climate Assessment and Data network (http://www.ecad.eu/) and were aggregated to average annual country estimates (Supplementary Figure 2). Forest structural data were available from Seidl et al.[38] and Vilén et al.[40] and included national estimates of total growing stock and median age at 5-year intervals (1985–2015; Supplementary Figure 3).

To temporally match both datasets as well as to focus the analysis on longer-term changes instead of year-to-year fluctuation, we averaged all data sets—including the annual mortality rates—to 5-year intervals, with the outer intervals including the first (1984) and last (2015) years (i.e., 1984–1989, 1990–1994, 1995–1999, 2000–2004, 2005–2009, 2010–2015). We subsequently regressed within-country mortality trends over trends in each covariate using Bayesian hierarchical log-linear models implemented in Stan via the rstanarm package (version 2.17.4)[65]. The model estimates the direction and strength of relationship between trends in mortality rates and trends in each covariate within each country, while accounting for random variation in intercept and slope between countries. We chose either a random intercept or a random intercept and random slope model based on the leave-one-out predictive performance measure implemented in the loo package (version 2.0.0)[66]. For further details on the modeling framework, we refer the reader to Supplementary Methods 3.

**Multi-proxy analysis of mortality change**. We compiled several additional proxies of forest mortality in order to facilitate a multi-proxy analysis of mortality change. We acquired data on the annual total round-wood volume harvested within each country from the FAOSTATS forest database (http://www.fao.org/faostat/). This dataset contains all wood removed from forests, including salvage logging in response to natural disturbances, and is reported by each country through annual surveys. We further obtained estimates of natural disturbances by wind and bark beetles covering the years 1984–2011 from Schelhaas et al.[33] and Seidl et al.[38]. Those estimates are based on gray literature reports of disturbance events in the region. Trends in wood removal and natural disturbances were estimated using Beta regression with a logit link function, implemented in Stan via the brms package (version 2.5.0)[67]. Trend estimates are thus comparable to our Landsat-based forest mortality change estimate. A tree-based analysis of forest mortality change was derived based on inventory data from the ICP network (http://icp-forests.net). Within the ICP network, forest health surveys are conducted annually at a $16 \times 16$ km$^2$ grid across the European continent[42]. The surveys were initiated in 1985, but starting dates vary across countries (see Supplementary Figure 4). At each plot, the defoliation status is assessed for at least 20 canopy trees per plot (no suppressed or intermediate trees; tree height >0.6 m). For our analysis, which focused on mortality rather than defoliation, we followed Neumann et al.[22] in assuming a tree to be dead if defoliation was 100% and the tree was no longer included in the assessment in the following year. We analyzed data of 77,592 trees across 2423 plots for the years 1990–2010 (the only period where data from all six countries were available; see Supplementary Figure 4), amounting to a total of 785,169 observations (trees × years). Individual tree mortality rates were calculated using the same model as for the Landsat time series analysis (see previous section). Hence, we estimated annual individual tree mortality rates and fractional changes in mean mortality rates over time for each country as well as jointly for all plots across the six countries.

**Code availability**. The model used for estimating annual rates and trends in mortality is available as R package under https://zenodo.org/record/1221340 [https://doi.org/10.5281/zenodo.1221339]. The complete analysis code is available under https://zenodo.org/record/1453351 [https://doi.org/10.5281/zenodo.1453350].

## Data availability

All data used in this study are deposited under https://zenodo.org/record/1453351 [https://doi.org/10.5281/zenodo.1453350]. A reporting summary for this article is available as a Supplementary Information file.

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

## Acknowledgements

C.S. acknowledges funding from the German Academic Exchange Service (DAAD) with funds from the German Federal Ministry of Education and Research (BMBF) and the People Programme (Marie Curie Actions) of the European Union's Seventh Framework Programme (FP7/2007–2013) under REA grant agreement Nr. 605728 (P.R.I.M.E. – Postdoctoral Researchers International Mobility Experience). R.S. and J.S. acknowledge support from the Austrian Science Fund FWF through START grant Y895-B25. We thank Dr. Warren Cohen and Dr. Steve Stehman for their valuable input during the design of this study. We acknowledge the data access and support provided by the Programme Coordinating Centre and the Expert Panels of ICP Forests. This analysis is partly based on data collected by partners of the official UNECE ICP Forests Network (http://icp-forests.net/contributors). Part of the data was financed by the European Commission. We finally acknowledge support by the German Research Foundation (DFG) through the Open Access Publication Fund of Humboldt-Universität zu Berlin.

## Author contributions

C.S., D.P., and R.S. designed the experiment. D.P. and Y.Z. implemented the TimeSync software. C.S., D.P., J.K., and J.S. collected the TimeSync data. M.N. coordinated the data access and processing of the ICP data. C.S. and R.S. analyzed the data. C.S and R.S. wrote the manuscript, with input from D.P., M.N., and P.H.

## Additional information

**Competing interests:** The authors declare no competing interests.

