## [Peer Review File · Nature Communications]

Reviewers' comments:

Reviewer #1 (Remarks to the Author):

Senf et al. present an analysis of forest mortality rates in six European countries, based primarily upon a large set of visually-interpreted Landsat timeseries. They find an increase in area-based mortality over the last 3 decades which is correlated to climate and growing stock, and then contrast this with other proxies of tree mortality. The introduction is clear, logical and well-situated in the literature, setting up the justification for the study very well. Two very important and novel things here are the length of the Landsat timeseries, and the consideration of non-stand-replacing mortality. The study provides another important piece of the story as to how mortality rates are changing across Europe, adding to previous work by Seidl et al. (2014, *Nature Climate Change* 4, 806-810) and Neumann et al. (2017, *Global Change Biology* 23, 4788-4797).

The area-based mortality rates from Landsat appear to be thoughtfully and robustly calculated. These are an important new data source that I hope will soon be published, and I salute the authors for the data-analysis effort involved! I also very much applaud the effort to try and carry out a multi-factor analysis, which is very much needed to help understand the drivers of changes in tree mortality rates, especially in such heavily-managed forests. The data here certainly possess the potential for a step forward in insight, although the analysis doesn't quite get to a clear attribution between demographic and climate drivers, which would be the holy grail here. I do, however, have some concerns about the way in which the multi-factor analysis is carried out and presented (much of the discussion hangs on the multi-factor aspect). I hope that the authors will be able to resolve these.

1) The comparison of changes in mortality rates between area-based mortality and stem mortality (lines 239-244) is oversimplified. These forests are increasing in biomass (as implied in Fig. 2a, although it would be nice to have the biomass trends also presented in the SI), however crown projective area and tree biomass do not scale linearly. Rather, self-thinning theory (e.g. Pretzsch, 2006, *Oecologia* 146, 572-583) indicates that biomass tends to increase more rapidly. Any comparison between these rates should take account of these allometric scaling relationships. The same argument applies to comparisons between area-based and stem mortality rates.

Is it indeed possible to reconcile the area-, stem- and volume-based metrics allometrically? Are trees sufficiently bigger that it is possible to maintain an increasing areal and volume loss rate with a decreasing stem loss rate (line 249)?

2) Likewise, the comparison of an area-based mortality rate to a volume-based (?) growth rate (line 285) needs appropriate correction.

3) The text presents a very strong link between the bark beetle disturbance timeseries and the areal-mortality rates (lines 225-229) which appears to be overstated. Whilst the logic of the mechanisms suggested is clear and reasonable, there is no clear trend in the presented bark beetle data (Fig. 4d). The rate of change is not statistically distinguishable from zero (Table S9), and further appears to be dominated by cyclic events with a wavelength of ca. 40% of the timeseries – i.e. within this timeseries it seems likely to be highly sensitive to the choice of start year. Whilst I'm very supportive of what the authors are trying to do and highly sympathetic to the problems of too-short timeseries for such problems, I don't think it is possible to draw any clear conclusions about the role of bark beetles in the increasing mortality rates from this data, and suggest that these sentences are removed from the discussion.

Indeed, the widespread increase in non-stand-replacing mortality could also be characteristic of direct drought stress. Perhaps it might also be worth to plot against a drought metric such as cumulative water deficit in Fig. 2?

4) Similarly, the ICP forests data shows a fairly constant mortality level for the last 14 or so years of the timeseries, with a large hump before that. It's impossible to say from this dataserie whether this hump is broadly characteristic of an earlier higher (and more variable) mortality rate, or an anomalous increase on a fairly constant background. Certainly, a linear fit does not appear appropriate for this data. I suggest the authors restrict their interpretation to a lack of evidence for

an increase in stem mortality rates, rather than a decrease. This need not necessarily change the conclusion that stem and areal mortality rates differ (but see above regarding allometric correction).

5) Additionally, is country-scale really the best aggregation level for correlations against temperature, and particularly precipitation? The countries here all have large topographic and climatic ranges. Would the correlations be stronger if more climatically-consistent sub-areas were used for correlations, before aggregating to country scale? ERA reanalyses at ca. 0.5° could be an alternative source of data for such a cross-check to avoid having to make decisions about such an aggregation.

Finally, as the authors note, there are many different ways in which mortality rate can be calculated, and the methods presented here do not all give the same direction of trend. The title appears therefore somewhat misleading, as what the authors are really reporting is that canopy projective area lost due to mortality has doubled over the last three decades – quite different from stem mortality rates, or biomass turnover rates. I suggest that the authors are careful throughout and in the title to refer to area affected by mortality when referring to the Landsat work, rather than just mortality (especially as they cover all main metrics of mortality within the manuscript), defining “affected” clearly in the abstract.

Further comments.

Fig. 1. The apparent drop in mortality rate since 2010 is interesting. Is this statistically significant? If so, can the authors comment on what it may signify? Possibly a reduction in size-based mortality rate increases in recent years in Austria and Germany? Seems to be opposite to the temperature trend?

Fig. 2. Can the authors separate a causal link between mortality and temperature from a link between temperature and growing stock, and thus mortality? How strong is the correlation between temperature and growing stock?

Lines 69-70. Inconclusive trends in mortality are not the reason for the significant uncertainty in forest model predictions, rather this is an uncertainty in mechanisms which clear trends alone would not resolve. Suggest to rephrase.

Line 92. For non-stand replacing mortality, is the lost forest area defined as the portion of the landsat pixel which was affected, or as the whole pixel (i.e. that pixel “experienced” mortality)? If the latter, to what extent does this mean that actual areal mortality rates are overestimated?

Results. It would help to define the meaning alpha and beta1 parameters in the main text when they are first introduced, rather than the reader having to skip to the Methods to interpret.

Line 171. Poland does not appear to show “much stronger” stand-replacing mortality increases than non-stand-replacing in Fig. 3. They appear to be statistically indistinguishable, and certainly not as strong as the opposite difference in the Czech Republic or Germany.

Line 184. Fig. 4 does not have correlation plots, however it would be useful to add these, perhaps as extra small panels to the right of the main timeseries for a vs b-d.

Line 207. This is certainly a wide-reaching and respect-worthy effort to draw together the available information on tree mortality rate changes in Europe. But in what sense is it comprehensive? In terms of available data? In my view, comprehensive would also include sub-canopy trees. It's not a problem that these are not included, of course, but perhaps some subtle rephrasing?

Line 215. Needs some references or data to support this point regarding the variability in management systems and forest types between countries.

Line 251. Also mention that it is consistent with mortality rate vs tree size relationships, e.g. Hülsmann et al. (2017)?

Line 282. Not clear here what “low to moderate severity” really means. Severe in terms of area affected during an event? Or in terms of % of trees in a given stand killed? Clarify or remove?

Line 314. It might be helpful for others if the result of this testing can be presented in the SI to facilitate informed decisions for future analyses.

Line 334. No Fig. S3? S2?

Reviewer #2 (Remarks to the Author):

This paper presents the first regionally consistent study of trends in forest disturbance in six European countries. The remote sensing methods are first rate, and the statistical estimation component is completely defensible. The paper actually downplays a significant technical achievement in using the ESA and USGS Landsat archives together. Historical formatting differences have prevented mining the two largely non-overlapping sets of imagery within the same application; this is the first large-scale use of a harmonized dataset.

The writing is clear, and the organization is reasonable. There is one major element of the paper that I think should be clarified, if not re-framed. The authors used a Landsat interpretation tool called TimeSync to estimate disturbance area using 30-m sample units. Having chosen a simple random sample of 30-m pixels in each country, it was fairly straightforward to turn the number of sample pixels with observed disturbance into a population estimates (with uncertainty) of area disturbed per year per country. The estimated area of disturbance generally rises from 1985 to 2016, which seemingly contradicts (ground) inventory estimates of declining numbers of trees dying. There is considerable discussion of why this might be so, but little discussion (save a too-brief allowance on line 269) of the fact that TimeSync is not actually capturing ALL mortality. TimeSync captures mortality events, where enough trees die within a 30-m pixel to perceptibly change the spectral signal. Given a standard density / stocking function typically used in forestry, trees die all the time, everywhere, when healthy stands develop (i.e. stocking goes up). With that in mind, perhaps the “tree level” and “area level” results are not contradictory at all: climatic stress could both increase the incidence of lots of trees dying in places that reach some physiological tipping point while at the same time decreasing everywhere else the self-thinning activity characteristic of a healthy stand.

The above is just my own conjecture; the authors are free to make their own interpretations. More important, though, is the fact that most of the paper talks about trends in mortality without emphasizing that TimeSync does not capture “healthy” mortality that occurs when some trees in a stand become dominant causing others to die. My conjecture above highlights why it is important to clarify that this paper focuses on mortality events (probably the same as “disturbance”) rather than just mortality. I feel that straightening this out throughout the manuscript to be a vital, but achievable, improvement before the paper is acceptable for this journal. I do recognize that some finesse will be required to define what a mortality event is, biophysically.

Some minor comments:

- Reference to growth rates in the Abstract (line 29) is confusing since the paper does not deal with measuring growth. I see the reference later, but perhaps the Abstract could specify that your comparison uses growth rates estimated by others
- Line 114 – not all studies show increasing severity. Cohen (2016, Forest Ecology and Management), in a study somewhat parallel to this one, used TimeSync to show the increasing incidence of low-magnitude disturbance
- Line 238 – Good point about the age variable meaning less in an era of retention forestry.

- Line 241 – Couldn't the difference in the two rates you mention be entirely explained by a shift to retention forestry? Why does it necessarily point to trends in non-management trends? Why not just say something like "natural processes are likely also at work"?
- Line 259-260 (among several others) – this statement is an example where it would be important to specify that you're measuring mortality events, not mortality.
- Better axis labels would benefit Figures S5 and S6

Reviewer #3 (Remarks to the Author):

Overall comments.

This paper addresses an important issue using a novel technique, and thus holds promise to contribute to the broader discussion of forest dynamics under a changing climate. As it presently stands, the analysis has two important limitations that appear to substantially weaken the authors' claims. It is possible that these can be addressed by a slight change in analysis, and by clarification of a key methodological step.

Issue 1: Discerning mortality using satellite imagery. The authors use a well-known tool to interpret forest change in a time series of satellite imagery. While I agree that forest mortality can often be interpreted using this technique, it is not clear that other phenomena might not also cause the spectral changes attributed to mortality. For example, ephemeral defoliation events last >1 year can cause a decline in leaf area, but not mortality; at the least, this effect needs to be considered in the interpretation.

More importantly, the distinction between events that are low vs high severity is important in the attribution section of the paper, coloring the interpretation of several of the patterns over time (particularly in relation to causation). That distinction is described in the supplemental material as being defined at vertices of the time series using a 10% forest cover cutoff. However, how is that call made? I am not aware that forest cover estimates can be made that precisely using the 30m pixels alone. Are air photos used? How? We need much more detail to have confidence that this number makes sense.

Issue 2: Mortality and temperature. The authors are right to examine relationships between mortality and factors that might cause tree stress such as increased temperature. After finding a relationship between temperature and mortality, the authors repeatedly state that increasing temperatures cause higher mortality. However, the implication is that increasing temperature in a given location would cause increased mortality at that location, but the test done in this analysis convolves geography and temperature change, making it impossible to test this assertion. In the current analysis, we can essentially know that countries with higher temperatures have higher mortality. But as I note in my detailed comments, this does not necessarily mean that mortality increased within a given country when temperature increased in that country.

I suggest a simple means of re-doing this analysis that may make the temperature control more direct.

Even if that comes up useful, it seems that we'd need to somehow control for increased harvest, since the authors make no claim about temperature leading to greater timber harvest. However, given that there is no way to separate harvest from non-harvest using severity alone, it seems impossible to fully separate these. Perhaps clues in the temporal dimension of the spectral signal, such as whether a spectral decline occurs over multiple years?

More generally, I am supportive of the mixture of sources to evaluate the trends noted over space and time. However, the interpretation seems to not match the evidence entirely. For example, if temperature is causing more mortality, would we not expect to see a commensurate trend in the

individual tree mortality data? The description of that discrepancy is not convincing. Similarly, it is an important issue that bark beetle outbreaks noted in the independent data set do not seem to be captured in the area-based estimates. I could see a variety of reasons why this would be the case, but this discrepancy is not fully explained.

Detailed notes (referenced by line number):

53-58. This sequence is not convincing. The first sentence in this section (about altering ecosystem structure and function) suggests to the reader that we're going to get some proof about the impact of tree mortality on ecosystem services. However, the next line only talks about carbon storage — it seems that calling out some other potential impacts would be good: perhaps water/temperature regulation, habitat for sensitive species, recreation, etc. It may be sufficient to call back to these functions mentioned on lines 43-46.

76: change to: "quality of reporting frequently declines in earlier years"

76: This statement seems to require a citation to justify — is it really contentious?

87: change to "We present here.."

92-94: the method here becomes very important — is it really possible to ascertain mortality from satellite imagery alone? It seems that the spectral signal could decline ephemerally due to factors that reduce productivity for a period, but would not necessarily be related to mortality. Indeed, spectral data have been linked to photosynthetic output for many global models. While I completely agree that mortality events would cause a decline, I'm not convinced that non-mortality events couldn't also cause such a decline. Is there some way to make this argument cleaner?

Looking at the supplemental section, it is not at all clear how the mortality estimation can be certain. I appreciate that the approach noted does seem to respond to mortality, but Table S3 seems to provide us with no evidence that high vs low severity mortality is actually distinguished, or how to ascertain the proportion of mortality within a 30m pixel (as suggested by line 92-93).

Is it that the stand-replacing mortality is defined by the label in Table S3 for the vertex land cover? How is this quantified at a sub-pixel scale if the only data available are the satellite pixels? This seems to be a key issue to resolve.

104-115: The distinction among types of disturbance is not entirely convincing, though this may just be a matter of semantics. From the introductory material, it seems it is important to distinguish between anthropogenic and natural causes of mortality. Yet the use of relative severity to tease these apart is problematic. For example, some natural insect pests cause near total mortality, at least in areas of the western U.S. and Canada. Also, the fact that low-severity change could be ascribed to either host tree specific insect activity or silviculture seems to weaken the argument.

119: structure of this sentence is awkward.

133-135: The statistics are plausible, assuming that the mortality itself is real! (see earlier comments).

152-154: I agree that the relationship shown in Figure 2 for temperature is basically robust. However, the interpretation needs to be handled carefully. The statement here in line 152-153 can easily be interpreted to mean that if temperature in a given location rises, mortality will increase.

But that is not what is tested in this analysis — this analysis essentially looks at whether the geography of temperature is related to mortality. My sense is that the authors are interested in asking whether mortality might be expected to increase under climate change, but we can't quite say that with this analysis. Looking at the specific data points (kudos to the authors for including all by period, by the way), one could argue that this analysis simply shows that warmer countries have more mortality than colder ones. In certain countries, it seems that the progression of temperature over time also relates to change in mortality, within country: poland, czechia. But in others (slovakia, perhaps austria [a little hard to tell colors]), it seems not. The challenge in interpretation, then, is whether the changes at the country level are related to temperature, or perhaps idiosyncrasies of historical or current management in each country. As the authors show later, Poland, for example, appeared to show a change in management strategy.

At the least, it would seem the authors should evaluate whether increase in temperature, normalized to the mean by country, also leads to an increase in mortality, normalized to the mean by country. This would make a much more convincing test of the temperature hypothesis.

192 — change “well-visible” to “evident”

215-216 — Agreed that there is consistent increase, but the results here suggest that the mechanism is quite different across countries. Also, on my first read through, I immediately wondered why Europe-wide patterns in economic activity are not listed — but these show up in the second factor (lines 230 forward), so it might be useful for the general reader if you can at least mention both initially, then delve into each separately.

218-220 - Again, the test here was not whether mortality increases as the temperature in a given location increases, which seems to be the implication of this sentence. This paragraph implies that the increase in mortality is associated *temporally* with the increase in temperature, but we can't say that for sure here.

Moreover, the results from other parts of the paper suggest that forest harvest is in fact a big driver of the overall increase in mortality. There doesn't seem to be a reasonable mechanism to link temperature to harvest.

22-224: Agreed that bark beetle reproduction and temperature are related, but the specific linkage between bark beetles and mortality seems weak here at best. By and large, it appears forest harvest is a big driver of the temporal trends. Moreover, since it's difficult to connect the lower-severity change exclusively to biotic agents, this claim is difficult to stand by.

230-239- this analysis seems to make sense.

241-243: I am not sure I understand this. It seems that the discrepancy could also be due to harvest happening in stands with lower growing stock, or?

249: How does this analysis support the notion that bigger trees are dying? I completely agree that big trees are important for the carbon story, but I need more direct help connecting that story to your own results.

255-256. Yes, this is a good recommendation.

267-272: While this sentiment is appropriate, the results presented here do not fully support it. The remote sensing tool is useful in being consistent over space, and largely over time, but the connection to climate seems tenuous (see prior comments) and the conflicting stories told by the proxies do not appear to tell an interpretable narrative here.

Reviewer 1

#	Comment	Response
1	Senf et al. present an analysis of forest mortality rates in six European countries, based primarily upon a large set of visually-interpreted Landsat timeseries. They find an increase in area-based mortality over the last 3 decades which is correlated to climate and growing stock, and then contrast this with other proxies of tree mortality. The introduction is clear, logical and well-situated in the literature, setting up the justification for the study very well. Two very important and novel things here are the length of the Landsat timeseries, and the consideration of non-stand-replacing mortality. The study provides another important piece of the story as to how mortality rates are changing across Europe, adding to previous work by Seidl et al. (2014, Nature Climate Change 4, 806-810) and Neumann et al. (2017, Global Change Biology 23, 4788-4797). The area-based mortality rates from Landsat appear to be thoughtfully and robustly calculated. These are an important new data source that I hope will soon be published, and I salute the authors for the data-analysis effort involved! I also very much applaud the effort to try and carry out a multi-factor analysis, which is very much needed to help understand the drivers of changes in tree mortality rates, especially in such heavily-managed forests. The data here certainly possess the potential for a step forward in insight, although the analysis doesn't quite get to a clear attribution between demographic and climate drivers, which would be the holy grail here. I do, however, have some concerns about the way in which the multi-factor analysis is carried out and presented (much of the discussion hangs on the multi-factor aspect). I hope that the authors will be able to resolve these.	We thank the reviewer for the supportive and constructive comments on our manuscript.

2	The comparison of changes in mortality rates between area-based mortality and stem mortality (lines 239-244) is oversimplified. These forests are increasing in biomass (as implied in Fig. 2a, although it would be nice to have the biomass trends also presented in the SI), however crown projective area and tree biomass do not scale linearly. Rather, self-thinning theory (e.g. Pretzsch, 2006, Oecologia 146, 572-583) indicates that biomass tends to increase more rapidly. Any comparison between these rates should take account of these allometric scaling relationships. The same argument applies to comparisons between area-based and stem mortality rates. Is it indeed possible to reconcile the area-, stem- and volume-based metrics allometrically? Are trees sufficiently bigger that it is possible to maintain an increasing areal and volume loss rate with a decreasing stem loss rate (line 249)?	Thank you very much for this comment. We agree that our initial comparison was oversimplified. As suggested by the reviewer, we now consider a more theoretical view on the relationship between canopy mortality, volume mortality and stem mortality in order to deduce expectations regarding their relationships. We explain this in detail in the Supporting Information (see Supplement S10), and have revised the introduction and discussion sections accordingly. In particular, we changed the last paragraph of the introduction to (L. 119): “Finally, we contextualized our results in a multi-proxy analysis of mortality indicators. Theory on change detection in ecosystems suggests that the inferential potential can be substantially increased by jointly studying multiple indicators of change⁴³. Hence, in addition to remotely sensed canopy mortality changes, we also analyzed trends in wood removal from official harvesting reports⁴⁴, individual tree mortality estimates from 785,169 tree-level observations of the ICP forest network^{22,45}, and grey literature estimates of changes in the most important abiotic (wind) and biotic (bark beetles) disturbance agents in the region³³. As these additional datasets report on specific aspects of forest mortality (e.g., wood removals, disturbance) rather than providing a comprehensive account of all mortality causes (as does our satellite-based estimate), a comparison across datasets is not suitable to validate our methodology. It rather provides a multi-faceted view of the changes in a crucial ecosystem process, and amends the remotely sensed information presented here with regard to its causal interpretation. Based on theoretical considerations on the relationship of canopy, volume and individual tree mortality in combination with increasing age and growing stock of Europe’s forests³⁹, we expected a stronger increase in canopy mortality than in mortality rate relative to growing stock, and a negative trend in individual tree mortality (see
---	--	--

		Supplement S10).” The revised discussion reads (L. 259): “Trends in mortality across our multi-proxy analysis were divergent, but generally conformed to theoretical expectations (see Supplement S10), with inventory-based individual tree mortality decreasing over past decades, while canopy mortality and official wood removal statistics indicating strongly increasing trends. Our multi-proxy analysis thus suggests that larger forest areas and/or areas of high growing stocks are particularly affected by mortality, with fewer (but bigger) trees dying in these forests. This pattern is consistent with changes in the demography of temperate forests in Europe^{39,40}, with empirical relationships between tree size and stem mortality⁴⁹, and with theoretical considerations (see Supplement S10). Moreover, it has important implications [...]”
3	Likewise, the comparison of an area-based mortality rate to a volume-based (?) growth rate (line 285) needs appropriate correction.	We agree that this comparison was overly naive. We have thus decided to omit such a direct comparison and rather note that mortality might cancel out increases in productivity, as also suggest by Reyer et al. 2017. We have revised the sentence to (L. 299): “A doubling of canopy mortality within 33 years – as observed here – constitutes a substantial change in ecosystem dynamics, and has the potential to cancel out simultaneously observed increases in tree growth^{26,59,}”
4	The text presents a very strong link between the bark beetle disturbance timeseries and the areal-mortality rates (lines 225-229) which appears to be overstated. Whilst the logic of the mechanisms suggested is clear and reasonable, there is no clear trend in the presented bark beetle data (Fig. 4d). The rate of change is not	We agree with the reviewer that our discussion of the role of bark beetle disturbances was not fully reflected in the presented data. While bark beetles certainly contribute, the magnitude of their contribution cannot be faithfully determined from our data. We thus followed the reviewers suggestion to omit such an interpretation, and

	statistically distinguishable from zero (Table S9), and further appears to be dominated by cyclic events with a wavelength of ca. 40% of the timeseries – i.e. within this timeseries it seems likely to be highly sensitive to the choice of start year. Whilst I’m very supportive of what the authors are trying to do and highly sympathetic to the problems of too-short timeseries for such problems, I don’t think it is possible to draw any clear conclusions about the role of bark beetles in the increasing mortality rates from this data, and suggest that these sentences are removed from the discussion. Indeed, the widespread increase in non-stand-replacing mortality could also be characteristic of direct drought stress. Perhaps it might also be worth to plot against a drought metric such as cumulative water deficit in Fig. 2?	have revised the section accordingly (L. 223): “The increase in canopy mortality was largely consistent across countries, despite their high variability in forest types and management systems”⁴⁶. This suggests that broader-scale processes such as climate change and forest recovery from past land-use – i.e. drivers affecting ecosystems dynamics across national borders and at large spatial scales – are important factors contributing to the observed increases in forest mortality. Our finding of a significant relationship of mortality with increasing temperature and growing stocks consistently across all countries supports this hypothesis. Both climate change (i.e., changing temperatures) and forest recovery (i.e., increase in growing stocks) are, for instance, important drivers of the prevailing natural disturbance regime in temperate Europe^{35,41}” While we agree that drought is an important driver of tree mortality, e.g. in the Mediterranean region of Europe, previous compilations of drought-related forest mortality (Allen et al. 2010) have not identified major events for the Central European countries under study here. Our finding of an only weak relationship of mortality with water availability (cf. Figure 2) confirms the subordinate role of drought in our study system, which is why we have refrained from conducting further analyses in this regard.
5	Similarly, the ICP forests data shows a fairly constant mortality level for the last 14 or so years of the timeseries, with a large hump before that. It’s impossible to say from this data series whether this hump is broadly characteristic of an earlier higher (and more variable) mortality rate, or an anomalous increase on a fairly constant background. Certainly, a linear fit does not appear appropriate for this data. I suggest the authors restrict their	Based on the reviewers comment we revisited our ICP analysis in order to better elucidate the origin and nature of the large hump referred to by the reader. In the course of this process we detected an error in our ICP database that resulted to a double-counting of dead trees that were not removed from the ICP plot protocols in the following year. This error caused the large hump in the beginning of the time series, and also biased the trend estimation reported in Table

	interpretation to a lack of evidence for an increase in stem mortality rates, rather than a decrease. This need not necessarily change the conclusion that stem and areal mortality rates differ (but see above regarding allometric correction).	S9. We have now corrected the error and – as expected by the reviewer – the significant trend disappeared (-1.45 [-4.31 - 1.36]). Thus, we followed the reviewers’ suggestion and report no evidence for changes in individual tree mortality. This finding is also more consistent with our theoretical expectations outlined in detail in the revised supporting information (see Supplement S10; see also answer to Comment #2). We revised the results section accordingly (L. 154ff). We are grateful to the reviewer for pointing us towards this issue, as it helped us to find and resolve an important issue in our database!
6	Additionally, is country-scale really the best aggregation level for correlations against temperature, and particularly precipitation? The countries here all have large topographic and climatic ranges. Would the correlations be stronger if more climatically-consistent sub-areas were used for correlations, before aggregating to country scale? ERA reanalyses at ca. 0.5° could be an alternative source of data for such a cross-check to avoid having to make decisions about such an aggregation.	We agree that the country scale is not the best aggregation scale for the climate driver analysis. However, we chose the country scale in order to be consistent with the structural variables, which are only available at the country scale. While combining our data with reanalysis datasets in order to further investigate the relationship between drought and forest mortality is certainly of great interest, we believe it is beyond the scope of this manuscript. A further methodological challenge of such an analysis would be the variable sampling density between countries. As is, our sampling design was optimized for analyses at the country scale, and corrections would be necessary if data were to be aggregated to another spatial unit, i.e., 0.5° cells.
7	Finally, as the authors note, there are many different ways in which mortality rate can be calculated, and the methods presented here do not all give the same direction of trend. The title appears therefore somewhat misleading, as what the authors are really reporting is that canopy projective area lost due to mortality has doubled over the last three decades – quite different from stem mortality rates, or biomass turnover rates. I suggest that the authors are careful throughout and in the title to refer to area affected by mortality	We agree with the reviewer that clear definitions are crucial and that our main results refer to canopy mortality. As this issue was also raised by other reviewers, we now refer to “canopy mortality” throughout the manuscript when referring to our Landsat-based mortality estimate. Accordingly, we also changed the title into: “Canopy mortality has doubled in Europe’s temperate forests over the last three decades”.

	when referring to the Landsat work, rather than just mortality (especially as they cover all main metrics of mortality within the manuscript), defining “affected” clearly in the abstract	
8	Fig. 1. The apparent drop in mortality rate since 2010 is interesting. Is this statistically significant? If so, can the authors comment on what it may signify? Possibly a reduction in size-based mortality rate increases in recent years in Austria and Germany? Seems to be opposite to the temperature trend?	This is an interesting point raised by the reviewer. While we are cautious in interpreting the significance of the trend after 2010 (only 7 years of data), we think that it can be explained by a ‘regression-to-the-mean’ phenomenon. That is, after the very high rates in the late 2000’s, there is a relatively high change of a decrease in mortality events towards the mean. Further, this behavior is consistent with fluctuating behavior of natural disturbances regimes identified for Europe, e.g. by Senf and Seidl (2018), where waves of high disturbances activity alternate with phases of low disturbance activity. Those waves were found to be linked to phases of high temperature/low precipitation in previous analyses. While our data do not allow for projection, it might be speculated that a ‘new wave’ of high disturbance activity can be expected for the upcoming years, following the very hot and partly dry conditions observed in recent years.
9	Fig. 2. Can the authors separate a causal link between mortality and temperature from a link between temperature and growing stock, and thus mortality? How strong is the correlation between temperature and growing stock?	As both growing stock and temperature increased across Europe over the last three decades, the variables are naturally correlated over the study period. The reviewer is thus correct that temperature might have facilitated tree growth (see Pretzsch et al. 2014) and thus mortality through higher susceptibility of mature trees to natural disturbances and management. While it was beyond the scope of our analysis to disentangle these complex interactions of potential drivers, Seidl et al. (2011) showed in a detailed attribution study using structural equation modeling that both growing stock change and temperature change influenced disturbance change even when accounting for their correlation.

10	Lines 69-70. Inconclusive trends in mortality are not the reason for the significant uncertainty in forest model predictions, rather this is an uncertainty in mechanisms which clear trends alone would not resolve. Suggest to rephrase.	We agree with the reviewer and have rephrased the sentence to (L. 72): “Consequently, the recent trends in forest mortality are inconclusive for many forest ecosystems around the globe. This lack of understanding is particularly worrying given that understanding mortality trends and their underlying drivers is an important prerequisite for improving predictions of future forest dynamics³⁴.”
11	Line 92. For non-stand replacing mortality, is the lost forest area defined as the portion of the landsat pixel which was affected, or as the whole pixel (i.e. that pixel “experienced” mortality)? If the latter, to what extent does this mean that actual areal mortality rates are overestimated?	A very good point raised by the reviewer. We do not correct for the proportion of the Landsat pixel which was affected. Thus, while there might be residual trees within the Landsat pixel, the total pixel is counted as experiencing a mortality event. For that reason, we explicitly define our estimates of mortality at the grain of Landsat pixels (see L. 96). As noted correctly by the reviewer, this also implies that simply multiplying the forest mortality rates with the total forest area within a country might lead to an overestimation of the total forest area at finer scales. Yet, as exact information on sub-pixel mortality rates are not available, we don’t see any possibility for correcting this.
12	Results. It would help to define the meaning alpha and beta parameters in the main text when they are first introduced, rather than the reader having to skip to the Methods to interpret.	We agree with the reviewer. We now define both alpha and beta when they first appear in the main text.
13	Line 171. Poland does not appear to show “much stronger” stand-replacing mortality increases than non-stand-replacing in Fig. 3. They appear to be statistically indistinguishable, and certainly not as strong as the opposite difference in the Czech Republic or Germany.	We agree with the reviewer. We rephrased the sentence to (L. 177): “An opposite pattern was observed for Poland, Austria, and Slovakia, which all showed increases in both stand-replacing and non-stand-replacing mortality.”
14	Line 184. Fig. 4 does not have correlation plots, however it would	We agree that correlation plots would add important information for

	be useful to add these, perhaps as extra small panels to the right of the main timeseries for a vs b-d.	the reader. We thus amended Figure 4 to also include correlation plots as suggested by the reviewer.
15	Line 207. This is certainly a wide-reaching and respect-worthy effort to draw together the available information on tree mortality rate changes in Europe. But in what sense is it comprehensive? In terms of available data? In my view, comprehensive would also include sub-canopy trees. It's not a problem that these are not included, of course, but perhaps some subtle rephrasing?	We agree with the reviewer that comprehensive is a very broad term that needs further specification here. As stated already in the introduction, “comprehensive” is here defined as “capturing the diverse set of prevailing causes of mortality“ (see introduction L. 89). To clarify this point for the reader, we repeat this notion also in the discussion in L. 215: “We here present the first consistent (across space and time) and comprehensive (i.e., capturing the diverse set of prevailing causes of mortality) assessment of canopy mortality across more than 30 Mill. ha of temperate forest in Europe.” In addition, the fact that we here do not consider mortality of sub-canopy trees is now made more obvious to the reader by rewording to “canopy mortality” throughout the text.
16	Line 215. Needs some references or data to support this point regarding the variability in management systems and forest types between countries.	We added a reference to support the statement.
17	Line 251. Also mention that it is consistent with mortality rate vs tree size relationships, e.g. Hülsmann et al. (2017)?	Good point. We included a reference to Hülsmann et al. 2017 indicating that our results are consistent with empirical relationships between tree size and stem mortality rates (L. 247): “This pattern is consistent with changes in the demography of temperate forests in Europe^{39,40}, with empirical relationships between tree size and stem mortality⁴⁹, and with theoretical considerations (see Supplement S10).”
18	Line 282. Not clear here what “low to moderate severity” really means. Severe in terms of area affected during an event? Or in	We agree with the reviewer that the sentence was unclear. We deleted the statement and rephrased the concluding paragraph to

	terms of % of trees in a given stand killed? Clarify or remove?	(L. 281): “Europe’s forest ecosystems are changing, and mortality is likely to be among the ecosystem processes changing most drastically. A doubling of canopy mortality within 33 years – as observed here – constitutes a substantial change in ecosystem dynamics, and has the potential to cancel out simultaneously observed increases in tree growth^{26,59}. The drastic changes in forest mortality documented here also highlight the importance of continuously monitoring ecosystem change, and utilize this information to foster forest resilience in policy and management decisions.”
19	Line 334. No Fig. S3? S2?	Good catch. Corrected.

Reviewer 2

#	Comment	Response
1	This paper presents the first regionally consistent study of trends in forest disturbance in six European countries. The remote sensing methods are first rate, and the statistical estimation component is completely defensible. The paper actually downplays a significant technical achievement in using the ESA and USGS Landsat archives together. Historical formatting differences have prevented mining the two largely non-overlapping sets of imagery within the same application; this is the first large-scale use of a harmonized dataset.	Thank you very much for your review and constructive feedback on our manuscript. We agree that combining the ESA and USGS archive was a big milestone for this work and we thus appreciate the positive feedback.
2	The writing is clear, and the organization is reasonable. There is one major element of the paper that I think should be clarified, if not re-framed. The authors used a Landsat interpretation tool called TimeSync to estimate disturbance area using 30-m sample units. Having chosen a simple random sample of 30-m pixels in each country, it was fairly straightforward to turn the number of sample pixels with observed disturbance into a population estimates (with uncertainty) of area disturbed per year per country. The estimated area of disturbance generally rises from 1985 to 2016, which seemingly contradicts (ground) inventory estimates of declining numbers of trees dying. There is considerable discussion of why this might be so, but little discussion (save a too-brief allowance on line 269) of the fact that TimeSync is not actually capturing ALL mortality. TimeSync captures mortality events, where enough trees die within a 30-m pixel to perceptibly change the spectral signal. Given a standard density / stocking function typically used in forestry, trees die all the time, everywhere, when healthy stands develop (i.e. stocking goes up). With that in mind, perhaps the “tree level” and “area level” results are not contradictory at all: climatic	Thank you very much for this comments. We agree with the reviewer that the results of increasing area-based mortality and decreasing stem-based mortality are not in fact contradictory. As this point was also raised by other reviewers, we added a theoretical explanation of our expectations into the supplement (Supplement S10), explaining why in stem-based mortality is expected to decrease (or stay stable), while area and volume based mortality estimates are expected to show an increasing trend in Europe’s forests. We also revised the introduction accordingly (L. 119): “Finally, we contextualized our results in a multi-proxy analysis of mortality indicators. Theory on change detection in ecosystems suggests that the inferential potential can be substantially increased by jointly studying multiple indicators of change⁴³. Hence, in addition to remotely sensed canopy mortality changes, we also analyzed trends in wood removal from official harvesting reports⁴⁴, individual tree mortality estimates from 785,169 tree-level observations of the ICP forest network^{22,45}, and grey literature estimates of changes in the most important abiotic (wind) and biotic

	stress could both increase the incidence of lots of trees dying in places that reach some physiological tipping point while at the same time decreasing everywhere else the self-thinning activity characteristic of a healthy stand. The above is just my own conjecture; the authors are free to make their own interpretations.	(bark beetles) disturbance agents in the region³³. As these additional datasets report on specific aspects of forest mortality (e.g., wood removals, disturbance) rather than providing a comprehensive account of all mortality causes (as does our satellite-based estimate), a comparison across datasets is not suitable to validate our methodology. It rather provides a multi-faceted view of the changes in a crucial ecosystem process, and amends the remotely sensed information presented here with regard to its causal interpretation. Based on theoretical considerations on the relationship of canopy, volume and individual tree mortality in combination with increasing age and growing stock of Europe's forests³⁹, we expected a stronger increase in canopy mortality than in mortality rate relative to growing stock, and a negative trend in individual tree mortality (see Supplement S10)." We also extended the discussion on how other satellite systems (i.e., space-borne lidar) might help to further understanding structural changes associated with canopy changes estimated by Landsat (L. 276): "Our analysis, for instance, only pertains to tree mortality in the canopy layer, underlining the need to complement remote sensing information with terrestrial data on forest health. Canopy-penetrating, active satellite systems might allow deeper insights into the structural changes associated with canopy mortality⁵⁸. Yet, those systems are currently limited to the analysis of recent mortality events, precluding the quantification of temporal trends." Finally we better explain that our estimate refer to canopy mortality, but see our response to the next comment.
3	More important, though, is the fact that most of the paper talks	We agree that our approach focuses on discrete mortality events, i.e.

	about trends in mortality without emphasizing that TimeSync does not capture “healthy” mortality that occurs when some trees in a stand become dominant causing others to die. My conjecture above highlights why it is important to clarify that this paper focuses on mortality events (probably the same as “disturbance”) rather than just mortality. I feel that straightening this out throughout the manuscript to be a vital, but achievable, improvement before the paper is acceptable for this journal. I do recognize that some finesse will be required to define what a mortality event is, biophysically.	both natural and anthropogenic disturbances in an ecological sense. We also agree that this point needs to be clarified throughout the manuscript. In order to clarify this in the revised version of the manuscript, we first changed the wording from forest mortality to canopy mortality when referring to our Landsat based estimate (see also Reviewer #1). Second, we revised our definition of canopy mortality (L. 93): “We visually interpreted 24,000 satellite-derived spectral time series spanning the period from 1984 to 2016 to quantify canopy mortality change across a forest area of approximately 30 Mill. ha, spanning six different countries in Europe (i.e., Austria, Czechia, Germany, Poland, Slovakia, and Switzerland). We defined canopy mortality rate as the percent forest area in which the dominant tree layer experienced a discrete mortality event (i.e., a disturbance from natural or anthropogenic causes) in a given year, assessed at a grain of 30 m pixels.”
4	Reference to growth rates in the Abstract (line 29) is confusing since the paper does not deal with measuring growth. I see the reference later, but perhaps the Abstract could specify that your comparison uses growth rates estimated by others	We agree with the reviewer and have removed the statement from the abstract.
5	Line 114 – not all studies show increasing severity. Cohen (2016, Forest Ecology and Management), in a study somewhat parallel to this one, used TimeSync to show the increasing incidence of low-magnitude disturbance	We agree that this statement was oversimplified and have now removed it from the manuscript.
6	Line 238 – Good point about the age variable meaning less in an era of retention forestry.	Thank you.
7	Line 241 – Couldn’t the difference in the two rates you mention be	We agree that our interpretation was not very clear in this point of

	entirely explained by a shift to retention forestry? Why does it necessarily point to trends in non-management trends? Why not just say something like “natural processes are likely also at work”?	the text. As this sentence was also criticized by other reviewers, we decided to drop it from the manuscript. Nonetheless, we think there is a more theoretical explanation for the higher rate in canopy mortality than in volume-based mortality, which we now outline in Supplement S10. Further, the volume-based estimate is based on reported wood-removal. As not all dead trees are removed from the forest, a divergence between trends can be expected. We revised the paragraph accordingly (L. 242): “Trends in mortality across our multi-proxy analysis were divergent, but generally conformed to theoretical expectations (see Supplement S10), with inventory-based individual tree mortality decreasing over past decades, while canopy mortality and official wood removal statistics indicating strongly increasing trends. Our multi-proxy analysis thus suggests that larger forest areas and/or areas of high growing stocks are particularly affected by mortality, with fewer (but bigger) trees dying in these forests. This pattern is consistent with changes in the demography of temperate forests in Europe^{39,40}, with empirical relationships between tree size and stem mortality⁴⁹, and with theoretical considerations (see Supplement S10). Moreover, it has [..]”
8	Line 259-260 (among several others) – this statement is an example where it would be important to specify that you’re measuring mortality events, not mortality.	Agreed. We revised the sentence to (L. 258): “However, as the increased canopy mortality observed in this study is strongly driven by intensified wood extraction, deadwood stocks are increasing at a much slower rate than the increase in mortality reported here would suggest⁵³.”
9	Better axis labels would benefit Figures S5 and S6	It is unclear to us what is meant by the reviewer. As the axis labels indicate the measurement and unit, we are unsure how to improve them.

Reviewer 3

#	Comment	Response
1	This paper addresses an important issue using a novel technique, and thus holds promise to contribute to the broader discussion of forest dynamics under a changing climate. As it presently stands, the analysis has two important limitations that appear to substantially weaken the authors' claims. It is possible that these can be addressed by a slight change in analysis, and by clarification of a key methodological step.	We very much thank the reviewer for this thorough review and the constructive suggestions on how to improve our work further. We carefully considered all the issues raised, and have implemented several substantial changes in the revised version of the manuscript, as well as we added additional analyses to support our conclusions. We feel that our work considerably benefited from these revisions and thank the Reviewer for suggestion them. Please see the following comments for further details.
2	Issue 1: Discerning mortality using satellite imagery. The authors use a well-known tool to interpret forest change in a time series of satellite imagery. While I agree that forest mortality can often be interpreted using this technique, it is not clear that other phenomena might not also cause the spectral changes attributed to mortality. For example, ephemeral defoliation events last >1 year can cause a decline in leaf area, but not mortality; at the least, this effect needs to be considered in the interpretation.	Please note that this answers also refers to comment #10. We thank the reviewer for this comment. We agree that also other phenomena than mortality can cause a change in the spectral signature. We here, however, undertook many steps towards ruling out those phenomena, such as minimizing residual clouds, cloud-shadows, accounting for differences in phenology at image acquisition, and for image misregistration. However, as the reviewer points out, also defoliation might cause spectral patterns similar to mortality, which might lead to false attribution of mortality to pixels that actually experience more ephemeral disturbances. While this is a valid concern in general, we feel that it is not a major issue in our analysis, and explain our rationale in detail in the following: First, there is only very limited large-scale insect defoliation across temperature forests of Europe. While insect defoliation is an issue in southern Europe (e.g., Pine processionary moth) and Northern Europe (e.g., geometrid moths), defoliating insects are largely absent from our study area, and do not form a substantial element of their natural disturbance regime. This is, of course, in strong contrast to,

		e.g., the Western US, where defoliators such as the western spruce budworm play an important role. A second reason for ephemeral defoliation could be air-pollution. However, a reduction in leaf area due to air-pollution is usually a rather localized phenomenon focused around particular emission sources (see Kandler and Innes 1995), and affects trees over much longer time horizons compared to the discrete mortality events recorded in our sample (90% are shorter than four years). Third, the interpreter mainly takes into account SWIR-based indices such as Tasseled Cap (TC) Wetness and the Normalized Burn Ratio (NBR) to determine a mortality event. Previous analyses showed that substantial changes in forest stand structure are required to result in a distinct signal in those indices. Those changes are normally associated with tree death. This contrasts NIR-based indices, such as the Normalized Difference Vegetation Index (NDVI), which is very sensitive to changes in photosynthetic activity, rather than structural changes caused by tree mortality. We agree that the NDVI, or similar indices, are thus less well suited for detecting tree mortality, as deviations can also be caused by more ephemeral changes such as defoliation. There is a large body of literature supporting our use of SWIR-based indices for detecting mortality-related changes in spectral trajectories (see for example Hais et al. 2009 and Senf et al. 2017). Further, based on our own research on defoliators, we know that there are particular difference in the spectral trajectories between mortality events and ephemeral defoliation (see Senf et al. 2015). In summary, ephemeral defoliation that does not lead to substantial tree mortality is highly unlikely to be detected in the Landsat spectral signature used in the current manuscript. Finally, we'd like to note that not only the spectral trajectory, but
--	--	--

		also Landsat imagery in Tasseled Cap space (visualized via RGB, see Supplement Figure S13) and high-resolution imagery in Google Earth was taken into account to determine whether a pixel has experienced tree mortality or not. This concept of accumulating evidence from different sources has been introduced by Cohen et al. (2010), and was successfully applied in numerous studies already (e.g., Cohen et al. 2016, Potapov et al. 2015, Pflugmacher et al. 2012). Hence, the technique of manually distinguishing mortality from more ephemeral disturbances using SWIR-based spectral trajectories, Landsat images and high-resolution imagery has matured considerably in recent years. We are thus confident that using this method for estimating mortality rates and trends for temperate forests of Europe is appropriate. References cited in the response but not in the manuscript: Kandler and Innes 1995, Air pollution and forest decline in Central Europe, Environmental Pollution, 90 Hais et al. 2009, Comparison of two types of forest disturbance using multitemporal Landsat TM/ETM+ imagery and field vegetation data, Remote Sensing of Environment, 113 Senf, C., et al. 2017. Remote sensing of forest insect disturbances: Current state and future directions, International Journal of Applied Earth Observation and Geoinformation, 60 Senf et al. 2015, Characterizing spectral-temporal patterns of defoliator and bark beetle disturbances using Landsat time series, Remote Sensing of Environment, 170 Cohen et al. 2010, Detecting trends in forest disturbance and
--	--	---

		recovery using yearly Landsat time series: 2. TimeSync — Tools for calibration and validation, Remote Sensing of Environment, 114 Cohen et al. 2016, Forest disturbance across the conterminous United States from 1985-2012: The emerging dominance of forest decline, Forest Ecology and Management, 360 Potapov et al. 2015, Eastern Europe's forest cover dynamics from 1985 to 2012 quantified from the full Landsat archive, Remote Sensing of Environment, 159 Pflugmacher et al. 2012, Using Landsat-derived disturbance history (1972–2010) to predict current forest structure, Remote Sensing of Environment, 122
--	--	--

3	More importantly, the distinction between events that are low vs high severity is important in the attribution section of the paper, coloring the interpretation of several of the patterns over time (particularly in relation to causation). That distinction is described in the supplemental material as being defined at vertices of the time series using a 10% forest cover cutoff. However, how is that call made? I am not aware that forest cover estimates can be made that precisely using the 30m pixels alone. Are air photos used? How? We need much more detail to have confidence that this number makes sense.	Please note that this answers also refers to comment #10. We thank the reviewer for this comment. We agree that we did not well describe our distinction between high-severity stand-replacing (SR) and low-severity non-stand-replacing (NSR) mortality events. You are indeed right that we used the land cover information at the end vertex of each mortality segment to classify mortality events into SR and NSR mortality events. We describe this explicitly in the revised manuscript in L. 342: “[...] based on the land cover information recorded for each vertex, we distinguished stand-replacing mortality events (i.e., resulting in a non-treed land cover) from non-stand-replacing mortality events (i.e., resulting in a treed land cover).”. The 10% tree cover cutoff, used to separate between treed and non-treed land cover and mentioned in the original version of the text, originated from the FAO definition of forests, which was adopted in this study. However, we agree that estimating exact percentages of tree cover per pixel is challenging without high-resolution imagery. We approached this problem by calibrating each interpreter at plots where high-resolution data was available in Google Earth. In particular, the interpreter compared the Tasseled Cap images (displayed in RGB space; see Supplement Figure S13) with high-resolution imagery to calibrate his ability in judging whether a mortality event resulted in a loss of all trees (i.e., resulting in a non-tree vertex), or if residual trees were left on the pixel (i.e., resulting in a treed-vertex). Even if no high resolution imagery was available for a given mortality event, more recent imagery in Google Earth was consulted for making the call on whether it was a SR or NSR mortality event. In particular, if – several years after the mortality event – the forest cover is characterized by homogenous regeneration, the mortality event can be considered SR. In a similar fashion also the size and shape of the mortality event (i.e., clear-cuts, large-scale wind events with salvage logging) can help in attributing
---	---	---

the post-mortality land cover.

In order to better explain our approach, we extended the description of the methods in L. 340:

“We did not separate specific agents of canopy cover change, as no historic high-resolution imagery was available throughout all countries, potentially introducing a significant attribution bias in earlier years. However, based on the land cover information recorded for each vertex, we distinguished stand-replacing mortality events (i.e., resulting in a non-treed land cover) from non-stand-replacing mortality events (i.e., resulting in a treed land cover). Estimating the percent of forest cover at the level of a Landsat pixel is inherently difficult. We here used high resolution imagery to calibrate interpreters in making decisions on whether a mortality event results in a complete loss of forest cover (stand-replacing mortality), or whether residual live trees remained (non-stand-replacing mortality). We subsequently evaluated our classification of stand-replacing and non-stand-replacing mortality events by testing for differences in relative residual forest cover between both classes at 280 plot locations selected from an independent dataset (see Supplement S11 for details).”

However, as we agree with the Reviewer on the remaining uncertainties of our approach. To address this issue, we now conducted a new analysis to test our assumption that there are substantial differences in residual tree cover between our SR and NSR class. Specifically, we have added a validation analysis to the supplement (Supplement S10), comparing the forest cover estimates by Hansen et al. (2013) for the year 2000 between SR and NSR mortality events from the year 1998-2000 (n=280). We found that

		there were significant differences in residual forest cover between the two classes distinguished here, with NSR mortality event having an on average 40% higher residual forest cover than SR disturbance events. Hence, the test confirms that our analysis was able to pick up major differences in residual tree cover and thus mortality severity, which adds confidence to our classification. Details on the validation experiment can be found in Supplement S11.
4	Issue 2: Mortality and temperature. The authors are right to examine relationships between mortality and factors that might cause tree stress such as increased temperature. After finding a relationship between temperature and mortality, the authors repeatedly state that increasing temperatures cause higher mortality. However, the implication is that increasing temperature in a given location would	Please note that this answer also refers to comment #14 and #17. We thank the reviewer very much for this comment. As we noted from the comment, there was a misunderstanding of the method employed in our study to determine the influence of covariates such as temperature, which likely resulted from a sub-optimal description

cause increased mortality at that location, but the test done in this analysis convolves geography and temperature change, making it impossible to test this assertion. In the current analysis, we can essentially know that countries with higher temperatures have higher mortality. But as I note in my detailed comments, this does not necessarily mean that mortality increased within a given country when temperature increased in that country.

I suggest a simple means of re-doing this analysis that may make the temperature control more direct.

Even if that comes up useful, it seems that we'd need to somehow control for increased harvest, since the authors make no claim about temperature leading to greater timber harvest. However, given that there is no way to separate harvest from non-harvest using severity alone, it seems impossible to fully separate these. Perhaps clues in the temporal dimension of the spectral signal, such as whether a spectral decline occurs over multiple years?

of our methods in the initial version of the manuscript.

Our model indeed tests for the relationship of within-country mortality trends and within-country trends of each covariate, as advised by the reviewer. The estimates given in Fig. 2 thus indicate the average fractional change in mortality rate over time, given a one unit change (over time) in each covariate. However, the model accounts for random variations in the model intercept among countries, which is the mortality rate at zero for each covariate. In this way, the model allows for detecting relationships between mortality trends and covariates over time, without being affected by between-country variations that might affect within-country trends. If supported by the data, we also took into account random variation of the slope parameter (that is, the strength and direction of the relationship) among countries. Hence, while the model estimates a mean relationship, country-specific relationships (i.e., running the model individually for each country) can also be estimated. We showed these country-specific estimates in the Supplement in Figure S8.

We recognized that the description of the statistical procedure was not ideal for explaining the complexity of our model. We thus revised the methods description in the manuscript, aiming to now better explain the model. Further, we added a detailed mathematical description of the model into the Supplement (Supplement S12). The revised methods sections reads as follows (L. 378):

“To temporally match both data sets as well as to focus the analysis on longer-term changes instead of year-to-year fluctuation, we averaged all data sets – including the annual mortality rates – to five year intervals, with the outer intervals including the first (1984) and last (2015) years (i.e., 1984-1989, 1990-1994, 1995-1999, 2000-

		2004, 2005-2009, 2010-2015). We subsequently regressed within-country mortality trends over trends in each covariate using Bayesian hierarchical log-linear models implemented in Stan via the rstanarm package⁶⁶. The model estimates the direction and strength of relationship between trends in mortality rates and trends in each covariate within each country, while accounting for random variation in intercept and slope between countries. We chose either a random intercept or a random intercept and random slope model based on the leave-one-out predictive performance measure implemented in the loo package⁶⁷. For further details on the modeling framework we refer the reader to Supplement S12.” Regarding your concern that increasing temperature also increases harvest: Indeed, that is possible. However, our hypothesis was not that increasing temperature solely increase natural disturbances. Our hypothesis was that mortality in general would be sensitive to regional-scale factors, such as changes in temperate, as was outlined in the introduction (L. 107): “To further elucidate the role of regional-scale factors for changing mortality, we regressed mortality trends over important variables of climate and forest structure at the country scale⁴¹.”
5	More generally, I am supportive of the mixture of sources to evaluate the trends noted over space and time. However, the interpretation seems to not match the evidence entirely. For example, if temperature is causing more mortality, would we not expect to see a commensurate trend in the individual tree mortality data? The description of that discrepancy is not convincing. Similarly, it is an important issue that bark beetle outbreaks noted in the independent data set do not seem to be captured in the area-based estimates. I could see a variety of reasons why this would be	We agree with the reviewer that our explanation for the divergent trends across the multiple mortality indicators presented was not fully convincing. However, we disagree that the opposing trends identified in our study are contradictory to our hypothesis. Indeed, with forests across Europe aging and accumulating biomass (see Figure S6 and Ciais et al. 2008), we’d expect a stronger increase in canopy based mortality than volume based mortality, and a negative trend in stem based mortality. To better explain this, we have now added a detailed theoretical explanation of our expectations to the

the case, but this discrepancy is not fully explained.

Supplement (S10), and have further revised the introduction and discussion sections to clarify the issue. The revised introduction now is (L. 119):

“Finally, we contextualized our results in a multi-proxy analysis of mortality indicators. Theory on change detection in ecosystems suggests that the inferential potential can be substantially increased by jointly studying multiple indicators of change⁴³. Hence, in addition to remotely sensed canopy mortality changes, we also analyzed trends in wood removal from official harvesting reports⁴⁴, individual tree mortality estimates from 785,169 tree-level observations of the ICP forest network^{22,45}, and grey literature estimates of changes in the most important abiotic (wind) and biotic (bark beetles) disturbance agents in the region³³. As these additional datasets report on specific aspects of forest mortality (e.g., wood removals, disturbance) rather than providing a comprehensive account of all mortality causes (as does our satellite-based estimate), a comparison across datasets is not suitable to validate our methodology. It rather provides a multi-faceted view of the changes in a crucial ecosystem process, and amends the remotely sensed information presented here with regard to its causal interpretation. Based on theoretical considerations on the relationship of canopy, volume and individual tree mortality in combination with increasing age and growing stock of Europe’s forests³⁹, we expected a stronger increase in canopy mortality than in mortality rate relative to growing stock, and a negative trend in individual tree mortality (see Supplement S10).”

The revised discussion now is (L. 242):

“Trends in mortality across our multi-proxy analysis were divergent, but generally conformed to theoretical expectations (see Supplement

		S10), with inventory-based individual tree mortality decreasing over past decades, while canopy mortality and official wood removal statistics indicating strongly increasing trends. Our multi-proxy analysis thus suggests that larger forest areas and/or areas of high growing stocks are particularly affected by mortality, with fewer (but bigger) trees dying in these forests. This pattern is consistent with changes in the demography of temperate forests in Europe^{39,40}, with empirical relationships between tree size and stem mortality⁴⁹, and with theoretical considerations (see Supplement S10). Moreover, it has important implications for the carbon cycling [...]
6	53-58. This sequence is not convincing. The first sentence in this section (about altering ecosystem structure and function) suggests to the reader that we're going to get some proof about the impact of tree mortality on ecosystem services. However, the next line only talks about carbon storage — it seems that calling out some other potential impacts would be good: perhaps water/temperature regulation, habitat for sensitive species, recreation, etc. It may be sufficient to call back to these functions mentioned on lines 43-46.	We agree that the flow of the paragraph was sub-optimal. We rephrased the paragraph as suggested to increase flow and logical consistency (L. 53): “However, elevated levels of tree death can substantially alter ecosystem structure and functioning, and impact the manifold services forest ecosystems provide to humanity^{9,10}. For example, increased mortality can impact drinking water quality¹¹ and timber supply¹². Moreover, elevated mortality decreases the carbon residence time in live biomass and soils^{13,14}, and could thus substantially reduce the carbon storage potential of forests¹⁵. Hence, increasing mortality rates are an important indicator of degrading forest health, which in turn could have strong detrimental effects on society^{16,17}.”
7	76: change to: “quality of reporting frequently declines in earlier years”	Changed as suggested.
8	76: This statement seems to require a citation to justify — is it really contentious?	We agree with the reviewer that “contentious” was not an ideal choice of word. However, as noted in Schelhaas et al. 2003, many uncertainties in trend detection stem from changing reporting quality

		over time. We rephrased the sentence and added Schelhaas et al. 2003 as reference (L. 80): “These datasets can provide large spatial coverage, but the quality of reporting frequently declines in earlier years, which makes change detection uncertain³³.”
9	87: change to “We present here..”	Changed to “We present [...]”.
10	92-94: the method here becomes very important — is it really possible to ascertain mortality from satellite imagery alone? It seems that the spectral signal could decline ephemerally due to factors that reduce productivity for a period, but would not necessarily be related to mortality. Indeed, spectral data have been linked to photosynthetic output for many global models. While I completely agree that mortality events would cause a decline, I’m not convinced that non-mortality events couldn’t also cause such a decline. Is there some way to make this argument cleaner? Looking at the supplemental section, it is not at all clear how the mortality estimation can be certain. I appreciate that the approach noted does seem to respond to mortality, but Table S3 seems to provide us with no evidence that high vs low severity mortality is actually distinguished, or how to ascertain the proportion of mortality within a 30m pixel (as suggested by line 92-93). Is it that the stand-replacing mortality is defined by the label in Table S3 for the vertex land cover? How is this quantified at a sub-pixel scale if the only data available are the satellite pixels? This seems to be a key issue to resolve.	Regarding your comment separating mortality from ephemeral disturbances, please see our response to comment #2 above. Regarding your comment on separating stand-replacing and non-stand-replacing mortality events, please see our response to comment #3 above.
11	104-115: The distinction among types of disturbance is not entirely convincing, though this may just be a matter of semantics. From the introductory material, it seems it is important to distinguish between	Thank you for this comment. First of all, our aim was not to distinguish between natural and anthropogenic mortality using the classification into high- and low-severity disturbances. We fully

	anthropogenic and natural causes of mortality. Yet the use of relative severity to tease these apart is problematic. For example, some natural insect pests cause near total mortality, at least in areas of the western U.S. and Canada. Also, the fact that low-severity change could be ascribed to either host tree specific insect activity or silviculture seems to weaken the argument.	agree that management as well as natural disturbance can both have high or low severity. Our aim rather was to test whether it is high intensity harvest and disturbances that affect large continuous patches (i.e., cyclonic wind events) which are increasing, or small-scale forest operations and selective biotic disturbances, which normally lead to much smaller disturbance patches and single tree mortality, as they only affect specific host trees. While we agree that biotic disturbance can cause large-scale patterns of tree mortality (especially in the US and Canada), this is rarely the case for Central Europe. The by far largest documented outbreak in Europe, located in the Bohemian Forest ecosystem (see Senf et al. 2017, Using Landsat time series for characterizing forest disturbance dynamics in the coupled human and natural systems of Central Europe, ISPRS Journal of Photogrammetry and Remote Sensing, Vol. 130), was several magnitudes smaller than recent insect outbreaks in the US or Canada. It further led to patchier disturbance patterns, as host trees (Picea in this case) are frequently mixed with non-host species (mostly Fagus in this case). But see also Senf and Seidl (2018) for an analysis of the spatial patterns of natural disturbances in Europe.
12	119: structure of this sentence is awkward.	Agreed. The sentence was rephrased to (L. 121): “Hence, in addition to remotely sensed canopy mortality changes, we also analyzed trends in wood removal from official harvesting reports⁴⁴, individual tree mortality estimates from 785,169 tree-level observations of the ICP forest network^{22,45}, and grey literature estimates of changes in the most important abiotic (wind) and biotic (bark beetles) disturbance agents in the region³³.”
13	133-135: The statistics are plausible, assuming that the mortality itself is real! (see earlier comments).	Please see our response to comments #2 and #3 above.

14	152-154: I agree that the relationship shown in Figure 2 for temperature is basically robust. However, the interpretation needs to be handled carefully. The statement here in line 152-153 can easily be interpreted to mean that if temperature in a given location rises, mortality will increase. But that is not what is tested in this analysis — this analysis essentially looks at whether the geography of temperature is related to mortality. My sense is that the authors are interested in asking whether mortality might be expected to increase under climate change, but we can't quite say that with this analysis. Looking at the specific data points (kudos to the authors for including all by period, by the way), one could argue that this analysis simply shows that warmer countries have more mortality than colder ones. In certain countries, it seems that the progression of temperature over time also relates to change in mortality, within country: poland, czechia. But in others (slovakia, perhaps austria [a little hard to tell colors]), it seems not. The challenge in interpretation, then, is whether the changes at the country level are related to temperature, or perhaps idiosyncrasies of historical or current management in each country. As the authors show later, Poland, for example, appeared to show a change in management strategy. At the least, it would seem the authors should evaluate whether increase in temperature, normalized to the mean by country, also leads to an increase in mortality, normalized to the mean by country. This would make a much more convincing test of the temperature hypothesis.	Please see our response to comment #4 above.
15	192 — change “well-visible” to “evident”	Changed as suggested.
16	215-216 — Agreed that there is consistent increase, but the results here suggest that the mechanism is quite different across countries.	While we see the point raised by the reviewer, we decided against mentioning both points in the first paragraph, as this would interrupt

	Also, on my first read through, I immediately wondered why Europe-wide patterns in economic activity are not listed — but these show up in the second factor (lines 230 forward), so it might be useful for the general reader if you can at least mention both initially, then delve into each separately.	the logical flow of the paragraph. Further, as the paragraph has been shortened substantially (see answer to comment #18), we think it is easier to follow our two arguments in the revised version of the manuscript.
17	218-220 - Again, the test here was not whether mortality increases as the temperature in a given location increases, which seems to be the implication of this sentence. This paragraph implies that the increase in mortality is associated *temporally* with the increase in temperature, but we can't say that for sure here. Moreover, the results from other parts of the paper suggest that forest harvest is in fact a big driver of the overall increase in mortality. There doesn't seem to be a reasonable mechanism to link temperature to harvest.	Please see our response to comment #4 above.
18	22-224: Agreed that bark beetle reproduction and temperature are related, but the specific linkage between bark beetles and mortality seems weak here at best. By and large, it appears forest harvest is a big driver of the temporal trends. Moreover, since it's difficult to connect the lower-severity change exclusively to biotic agents, this claim is difficult to stand by.	We agree that we somewhat over-stretched the interpretation of our data with regard to the potential role of biotic disturbances. We have now revised the discussion and abstract, keeping the interpretation more closely to the actual data (L. 223): “The increase in canopy mortality was largely consistent across countries, despite their high variability in forest types and management systems⁴⁶. This suggests that broader-scale processes such as climate change and forest recovery from past land-use – i.e. drivers affecting ecosystems dynamics across national borders and at large spatial scales – are important factors contributing to the observed increases in forest mortality. Our finding of a significant relationship of mortality with increasing temperature and growing stocks consistently across all countries supports this hypothesis. Both climate change (i.e., changing temperatures) and forest recovery

		(i.e., increase in growing stocks) are, for instance, important drivers of the prevailing natural disturbance regime in temperate Europe 35,41 ”
19	230-239- this analysis seems to make sense.	We thank the reviewer for this assessment.
20	241-243: I am not sure I understand this. It seems that the discrepancy could also be due to harvest happening in stands with lower growing stock, or?	We agree that this point was unclear in the previous version of the manuscript. As it was also highlighted by other reviewers, we decided to drop this last part from the paragraph in the revised version of the text.
21	249: How does this analysis support the notion that bigger trees are dying? I completely agree that big trees are important for the carbon story, but I need more direct help connecting that story to your own results.	An increase in canopy- and volume-based mortality with a simultaneous decrease in stem-based mortality can only be explained by fewer but larger trees dying. Indeed, our multi-proxy analysis is well in line with theoretical considerations, which we now outline in detail in the revised version of the supplement (Supplement S10).
22	255-256. Yes, this is a good recommendation.	We thank the reviewer for this appraisal.
23	267-272: While this sentiment is appropriate, the results presented here do not fully support it. The remote sensing tool is useful in being consistent over space, and largely over time, but the connection to climate seems tenuous (see prior comments) and the conflicting stories told by the proxies do not appear to tell an interpretable narrative here.	Please see our responses to comment #2, #3 and #4 above.

REVIEWERS' COMMENTS:

Reviewer #1 (Remarks to the Author):

I am satisfied with the authors' responses to my comments and I would like to congratulate them on a very nice analysis and an important contribution.

Just one further minor suggestion:

L232. I'm not sure if harvesting is a "second factor" contributing to the observed increase in canopy mortality, so much as a direct result of the first factor – larger and more mature trees are likely to result in more harvest. This link could be made more explicit and the confounding factor of salvage logging also mentioned.

Reviewer #2 (Remarks to the Author):

I am satisfied that the issues raised by my original review, most of which were also raised by other reviewers, have been addressed. My comments about Figures S5 and S6 related to labeling the y-axes as "Value." I realize that the variables and units are defined at the top of the figures, but if we take the trouble to write "value," couldn't we just repeat "precipitation (mm)"?

Reviewer #3 (Remarks to the Author):

I read with great interest the revised paper and the authors' responses to reviewer comments.

In the first version, I had concerns about the method used to determine mortality, and on the interpretation of temperature/mortality relationship within and among countries.

Reading the revised manuscript and the authors' responses, my concerns are alleviated. I believe the paper is methodologically sound, and tells a clear story about an important process. Moreover, the authors are careful in their interpretations. I am satisfied that the paper should move forward.

Reviewer #1 (Remarks to the Author):

I am satisfied with the authors' responses to my comments and I would like to congratulate them on a very nice analysis and an important contribution.

Just one further minor suggestion:

L232. I'm not sure if harvesting is a "second factor" contributing to the observed increase in canopy mortality, so much as a direct result of the first factor – larger and more mature trees are likely to result in more harvest. This link could be made more explicit and the confounding factor of salvage logging also mentioned.

Response: We agree and revised the discussion section accordingly (L. 287ff).

“While increasing natural disturbances likely contributed to the observed mortality trend³⁸, land-use change and specifically intensified tree harvesting – including salvage logging of naturally disturbed areas – is the most important agent of canopy mortality in Europe's temperate forests. This interpretation is supported by a strong correlation of observed canopy mortality trends with reported wood removals (Figure 4), suggesting that trees are removed for human usage from most of the areas experiencing mortality. Our results indicate that increased extraction is happening primarily in the form of non-stand-replacing mortality (Figure 3). This suggests increased thinning activity and a transition from past clear-cut systems towards “close-to-nature” silviculture³⁴ and retention forestry⁴⁶ in the temperate forests of Europe. [...]”

Reviewer #2 (Remarks to the Author):

I am satisfied that the issues raised by my original review, most of which were also raised by other reviewers, have been addressed. My comments about Figures S5 and S6 related to labeling the y-axes as "Value." I realize that the variables and units are defined at the top of the figures, but if we take the trouble to write "value," couldn't we just repeat "precipitation (mm)"?

Response: We agree and changed the axis labeling in the SI.

Reviewer #3 (Remarks to the Author):

I read with great interest the revised paper and the authors' responses to reviewer comments.

In the first version, I had concerns about the method used to determine mortality, and on the interpretation of temperature/mortality relationship within and among countries.

Reading the revised manuscript and the authors' responses, my concerns are alleviated. I believe the paper is methodologically sound, and tells a clear story about an important process. Moreover, the authors are careful in their interpretations. I am satisfied that the paper should move forward.

Response: We thank the reviewer.